# STAR: Efficient Preference-based Reinforcement Learning via Dual Regularization

**Fengshuo Bai**[1,2,3]   **Rui Zhao**[4,*]  **Hongming Zhang**[5]  **Sijia Cui**[5]  **Shao Zhang**[1]

**Bo Xu**[5]    **Lei Han**[4,*]   **Ying Wen**[1]   **Yaodong Yang**[6,*]

[1]Shanghai Jiao Tong University [2]PKU-PsiBot Joint Lab [3]Zhongguancun Academy [4]Tencent
[5]National Key Laboratory of Cognition and Decision Intelligence for Complex Systems,
Institution of Automation, Chinese Academy of Sciences
[6]Institute for AI, Peking University

## Abstract

Preference-based reinforcement learning (PbRL) bypasses complex reward engineering by learning from human feedback. However, due to the high cost of obtaining feedback, PbRL typically relies on a limited set of preference-labeled samples. This data scarcity introduces two key inefficiencies: (1) the reward model overfits to the limited feedback, leading to poor generalization to unseen samples, and (2) the agent exploits the learned reward model, exacerbating overestimation of action values in temporal difference (TD) learning. To address these issues, we propose STAR, an efficient PbRL method that integrates preference margin regularization and policy regularization. Preference margin regularization mitigates overfitting by introducing a bounded margin in reward optimization, preventing excessive bias toward specific feedback. Policy regularization bootstraps a conservative estimate $\widehat{Q}$ from well-supported state-action pairs in the replay memory, reducing overestimation during policy learning. Experimental results show that STAR improves feedback efficiency, achieving 34.8% higher performance in online settings and 29.7% in offline settings compared to state-of-the-art methods. Ablation studies confirm that STAR facilitates more robust reward and value function learning. The videos of this project are released at https://sites.google.com/view/pbrl-star.

## 1 Introduction

Reinforcement learning (RL) [47, 5] has demonstrated remarkable success across various domains [1, 66, 69, 35, 56, 26]. However, its reliance on well-designed reward functions remains a significant challenge, as defining rewards for complex tasks is often difficult [59, 39, 62]. For example, in robotic manipulation, re-orienting a cube with a dexterous hand requires quantifying intermediate states, which is non-trivial [3, 4, 58, 70].

Preference-based reinforcement learning (PbRL) offers an alternative by replacing explicit reward functions with human preferences over agent trajectories [24, 37, 28, 73, 57]. PbRL follows an iterative process: (1) collecting human feedback to train a reward model and (2) optimizing a policy using this learned reward model. This iterative process aligns the agent with human-desired behaviors without the need for manually specified dense reward signals.

Despite its advantages, PbRL suffers from two fundamental inefficiencies when human feedback is limited. First, the reward model tends to overfit due to the small amount of preference data, leading to poor generalization [13, 36, 60, 61, 54]. Second, policy learning exacerbates this issue: TD-based optimization relies on the learned reward model, and errors in reward estimation, combined

---

[*]Correspondence to: yaodong.yang@pku.edu.cn, leihan.cs@gmail.com, rui.zhao.ml@gmail.com.

with overestimated Q-value updates, cause unstable learning [13, 28, 64, 19]. These challenges are illustrated in Figure 1, which visualizes how reward overfitting and Q-value overestimation degrade PbRL performance. Similar issues have been discussed in prior work [28, 30], highlighting the need for more feedback-efficient solutions. Addressing these inefficiencies is crucial for making PbRL viable in real-world applications, where extensive human feedback is impractical.

To address these challenges, we introduce STAR, a feedback-efficient PbRL algorithm that integrates two complementary techniques: human preference**S** margin regulariza**T**ion and policy regul**AR**ization. Preference regularization mitigates overfitting in reward learning by introducing a bounded margin in the optimization objective of standard preference learning, preventing the model from overly focusing on limited feedback data. Policy regularization reduces overestimation bias during RL training by estimating a conservative $\widehat{Q}$ value using only transitions stored in replay

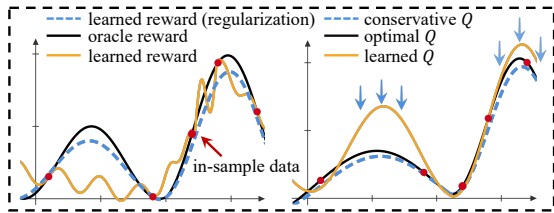

Figure 1: Illustration of two core inefficiencies—reward learning (left) and value estimation (right)—under limited human feedback. Our proposed STAR integrates preference regularization and conservative estimation to mitigate these challenges.

memory. This estimate is then used to constrain policy updates via Kullback-Leibler (KL) divergence, stabilizing learning. Our empirical evaluations across complex manipulation and locomotion tasks, in both online and offline PbRL settings, demonstrate that STAR consistently outperforms baselines. Notably, this advantage is particularly pronounced when human feedback is limited, highlighting STAR's ability to make efficient use of limited data for superior performance.

The key contributions of our work are as follows: (1) We propose STAR, a novel feedback-efficient PbRL algorithm that integrates preference margin regularization and policy regularization to improve learning stability and feedback utilization. (2) We demonstrate that STAR enhances feedback efficiency over existing PbRL methods across diverse online and offline tasks. Furthermore, human experiments reveal the emergence of novel behaviors driven by human feedback. (3) We provide in-depth analysis showing that preference margin regularization and policy regularization improve the accuracy of the reward model and mitigate Q-value overestimation, leading to improved policy optimization in feedback-limited settings.

## 2 Related Work

**Preference-based Reinforcement Learning.** PbRL is a novel way to learn agents from human feedback without reward engineering. Instead, a human provides preferences between the agent's behaviors, and the agent uses this feedback to perform the task. Christiano et al. [4] formulates a basic framework, and Ibarz et al. [17] utilizes imitation learning as the warm-start strategy to speed up PbRL. To further improve feedback efficiency, **PEBBLE** [24] replaces PPO [44] with SAC [11] to achieve data efficiency and combines unsupervised pre-training with the reward relabeling technique during learning. Building upon this foundation, Park et al. [37] introduces **SURF**, a semi-supervised reward learning framework that improves reward learning via pseudo-labels and temporal cropping augmentation. **MRN** [28] incorporates bi-level optimization for improving the quality of the Q function. **QPA** [16] improves feedback efficiency by addressing the issue of query-policy misalignment. Besides, some research has various considerations, such as skill extraction [55], intrinsic reward [27], meta-learning [14], and these methods have improved the efficiency to a certain extent. In addition to focusing on PbRL in online settings, a significant portion of research also concentrates on it in offline settings. **PT** [21] leverages a transformer architecture to learn a non-Markovian reward and preference weighting function. **IPL** [13] directly optimizes the implicit rewards deduced from the learned Q-function, ensuring alignment with expert preferences. Kang et al. [20] models offline trajectories and preferences in a one-step process without reward learning. **DTR** [51] balances staying close to the behavior policy with high-return trajectories and choosing actions with high reward labels. When dealing with language models, PbRL naturally facilitates the emergence of reinforcement learning from human feedback (RLHF) [36]. Before that, Stiennon et al. [45] have fine-tuned a summarizing policy following the PbRL paradigm. Our approach is orthogonal to previous approaches in that we use a conservative estimate $\widehat{Q}$ to regularize the neural Q-function to mitigate overestimation and leverage the preference margin regularization to prevent overfitting of the reward model.

## 3 Preliminaries

### 3.1 Preference-based Reinforcement Learning

In the preference-based RL framework, human preferences replace hand-craft rewards. Following the notation in Christiano et al. [4], a trajectory segment $\sigma$ is defined as a sequence of states and actions $(s_{t+1}, a_{t+1}, \ldots, s_{t+k}, a_{t+k})$. Given a pair of segments $(\sigma^0, \sigma^1)$, a human provides preference label $y$, which is a distribution over $\{(1,0), (0,1), (0.5, 0.5)\}$ indicating preference for $\sigma^0$, $\sigma^1$, or indifference.

The reward model $\widehat{r}_\psi$ assigns scores to state-action pairs, and the cumulative reward over a segment is used to predict human preferences. Following the Bradley-Terry model [2], the preference predictor is formulated as:

$$P_\psi[\sigma^0 \succ \sigma^1] = \text{Sigmoid}\Big( \sum_t \widehat{r}_\psi(s_t^0, a_t^0) - \sum_t \widehat{r}_\psi(s_t^1, a_t^1) \Big), \tag{1}$$

where $\sigma^0 \succ \sigma^1$ indicates $\sigma^0$ is more consistent with human expectations than $\sigma^1$.

The reward model is optimized by minimizing the cross-entropy loss between predicted and actual preferences:

$$\mathcal{L}_{\text{reward}}(\psi) = - \mathop{\mathbb{E}}_{(\sigma^0, \sigma^1, y) \sim \mathcal{D}_{\text{pref}}} \Big[ y(0) \log P_\psi[\sigma^0 \succ \sigma^1] + y(1) \log P_\psi[\sigma^1 \succ \sigma^0] \Big], \tag{2}$$

where $\mathcal{D}_{\text{pref}}$ denotes the dataset of preference pairs.

### 3.2 TD3-Based PbRL Framework

Twin Delayed Deep Deterministic Policy Gradient (TD3) [9] is an off-policy actor-critic algorithm designed to address value overestimation in deep reinforcement learning. When implementing PbRL with TD3, the learned reward model $\widehat{r}_\psi$ replaces the environment reward in the standard TD3 pipeline. The policy learning objective becomes maximizing :

$$\mathcal{J}_\pi(\phi) = \mathbb{E}_{s \sim \mathcal{D}} \left[ \min_{j=1,2} Q_{\theta_j}(s, \pi_\phi(s)) \right], \tag{3}$$

where $\mathcal{D}$ is the replay buffer of transition samples, and $Q_{\theta_j}$ is the $j$-th Q-function.

Each Q-function is trained to minimize the TD error with rewards from the learned model $\widehat{r}_\psi$:

$$\mathcal{J}_Q(\theta_j) = \mathbb{E}_{(s_t, a_t, s_{t+1}) \sim \mathcal{D}} \left[ \left( Q_{\theta_j}(s_t, a_t) - y_t \right)^2 \right], \tag{4}$$

where the target value $y_t = \widehat{r}_\psi(s_t, a_t) + \gamma \min_{j'=1,2} Q_{\bar{\theta}_{j'}}(s_{t+1}, \pi_{\bar{\phi}}(s_{t+1}) + \epsilon)$. Here, $\bar{\theta}$ and $\bar{\phi}$ denote the target networks for Q-functions and policy; $\gamma \in [0, 1)$ is the discount factor; and $\epsilon \sim \text{clip}(\mathcal{N}(0, \sigma), -c, c)$ is the clipped noise added for target policy smoothing.

## 4 Method

In this section, we introduce STAR, a flexible framework designed to enhance feedback efficiency in any PbRL approach. Figure 2 provides an overview of our method, which build up dual regularization through three core components: preference regularization, conservative value estimation $(\widehat{Q})$, and policy regularization. These components work together to optimize feedback utilization and improve policy performance. The detailed procedures for both the online and offline settings are outlined in Algorithms 1 and 2 in Appendix A.

### 4.1 Preference Regularization with Bounded Margin

**Intuition and Analysis.** We analyze the issue of reward over-optimization in preference-based learning. As discussed in Section 3.1, optimizing Equation (2) encourages the reward model to align with human preferences. However, this objective implicitly drives the optimization toward an unbounded reward difference between preferred and rejected segments: $\sum_t \widehat{r}_\psi(s_t^0, a_t^0) - \sum_t \widehat{r}_\psi(s_t^1, a_t^1) \to \infty$. Although the predicted rewards do not become literally infinite in practice, this unbounded optimization tendency indicates that Equation (2) is an ill-posed objective. It encourages the model to become overconfident in its predictions for training data, leading to poor generalization and the reward over-optimization problem. To address this, we introduce Preference Margin Regularization

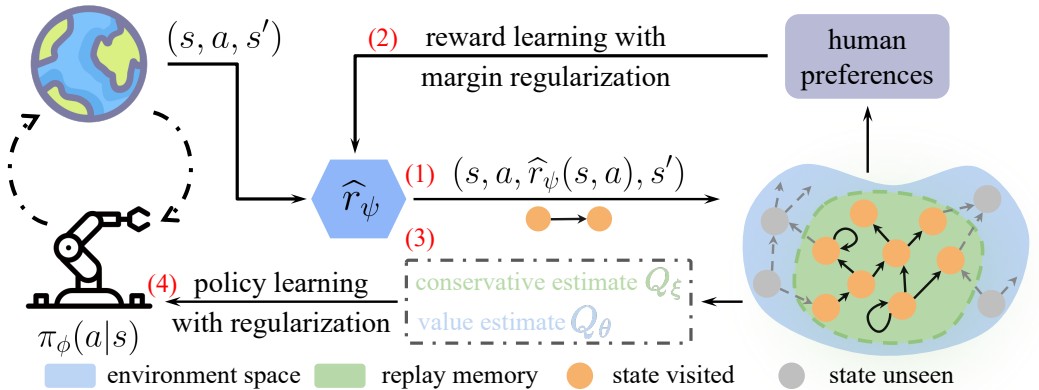

Figure 2: **Illustration of STAR.** (1) Label rewards using $\widehat{r}_\psi$. (2) Update the reward model $\widehat{r}_\psi$ via a bounded optimization objective from preference margin regularization. (3) Estimate conservative $Q_\xi$ and update $Q_\theta$. (4) Regularize $\pi_\phi$ with $Q_\xi$.

(PMR), which enforces a more reasonable decision boundary between preferred and rejected segments, thereby mitigating overfitting to training preferences. Detailed derivations and analyses are provided in Appendix B.1.

**PMR Formulation.** Drawing from margin learning theory [43, 46, 29], we propose PMR, which introduces a margin parameter $\varepsilon$ to smooth the target distribution:

$$y_{\text{smooth}} = (1 - \varepsilon) \cdot y + \varepsilon \cdot (1 - y). \tag{5}$$

The PMR loss function becomes:

$$\mathcal{L}_{\text{PMR}}(\psi) = - \mathbb{E}_{(\sigma^0, \sigma^1, y) \sim \mathcal{D}} \Big[ \sum_{i=0}^{1} y_{\text{smooth}}(i) \log P_\psi[\sigma^i \succ \sigma^{1-i}] \Big]. \tag{6}$$

By setting the derivative of this loss to zero and solving for the optimal solution, we can show that PMR implicitly enforces a bounded margin:

$$\sum_t \widehat{r}_\psi(s_t^0, a_t^0) - \sum_t \widehat{r}_\psi(s_t^1, a_t^1) = \log \left( \frac{1 - \varepsilon}{\varepsilon} \right). \tag{7}$$

This bounded margin between preferred and rejected samples replaces the unbounded optimization objective of standard preference learning, effectively preventing the reward model from becoming overly confident while maintaining sufficient discriminative power.

## 4.2 Policy Regularization with Conservative Q

Building upon our analysis of reward overfitting, we now address the complementary problem of Q-function overestimation in PbRL. Even with a well-calibrated reward model from PMR, standard Q-learning can still lead to overestimation, especially in regions with limited data.

**Conservative Estimate $Q_\xi$.** We introduce a conservative Q-function $Q_\xi$ that only utilizes well-supported state-action pairs from the current replay memory. This conservative estimation provides two main advantages: firstly, it enables further exploitation of samples in the replay memory; secondly, it prevents overestimation caused by extrapolation in unseen states and actions. Following principles from conservative Q-learning [23, 22, 10, 67, 68], we define a constrained operator $\mathcal{T}$:

$$\mathcal{T}^\beta Q_\xi(s_t, a_t) = \beta \log \mathbb{E}_{(s_t, a_t) \sim \mathcal{D}} \left[ \frac{1}{\beta} \exp(Q_\xi(s_t, a_t)) \right], \tag{8}$$

where $\beta > 0$ is a temperature parameter. Since state-action pairs are sampled from the replay buffer, the operator focuses exclusively on in-distribution actions. For any $\beta_1 > \beta_2 > 0$, we have $\mathcal{T}^{\beta_1} Q_\xi < \mathcal{T}^{\beta_2} Q_\xi$. As $\beta \to \infty$, $\mathcal{T}^\beta Q_\xi \to \mathbb{E}[Q_\xi]$, while $\beta \to 0$ recovers the in-distribution greedy estimate, i.e., $\mathcal{T}^\beta Q_\xi \to \sup(Q_\xi)$.

The conservative Q-function $Q_\xi$ is trained by minimizing:

$$\mathcal{J}_Q(\xi) = \mathbb{E}_{\tau_t^{\text{in}} \sim \mathcal{D}} \left[ \left( Q_\xi(s_t, a_t) - \widehat{r}_\psi(s_t, a_t) - \gamma \mathcal{T}^\beta Q_\xi(s_{t+1}, a_{t+1}) \right)^2 \right], \tag{9}$$

where in-distribution transition defined as: $\tau_t^{\text{in}} = (s_t, a_t, s_{t+1}, a_{t+1})$.

**Policy Regularization.** We leverage the conservative Q-function $Q_\xi$ to regularize the policy learning process. The key insight is to use $Q_\xi$ to derive a reference policy $\widehat{\pi}$ that captures the current best estimate of action distributions given uncertainty: $\widehat{\pi}(a|s) \propto \exp(Q_\xi(s, a))/Z(s)$, where $Z(s)$ is the partition function ensuring normalization.

We regularize the learned policy $\pi_\phi$ toward this reference policy using the KL divergence. Combined with the TD3 policy loss in Equation (3), the overall objective becomes:

$$\mathcal{J}_\pi(\phi) = \mathbb{E}_{s_t \sim \mathcal{D}} \left[ Q_\theta(s_t, \pi_\phi(s_t)) + \eta D_{\text{KL}} \left( \pi_\phi(\cdot|s_t) \| \widehat{\pi}(\cdot|s_t) \right) \right], \tag{10}$$

where $\eta > 0$ controls the regularization strength. This formulation ensures that the policy optimization is guided by both the current Q-function $Q_\theta$ and the conservative estimate $Q_\xi$.

### 4.3 Theoretical Analysis

We provide a theoretical analysis to show the properties of the conservative estimate, denoted as $\widehat{Q}$, in a finite state-action space $\mathcal{S} \times \mathcal{A}$. Unlike the standard Q-function $Q$, which bootstraps over the entire action space, $\widehat{Q}$ bootstraps only in-distribution actions, thereby reducing extrapolation error from out-of-distribution data. As a result, $\widehat{Q}$ provides a conservative (i.e., lower-bound) estimate of $Q$, and converges to the optimal value as data coverage increases. The proof of the following theorem is provided in Appendix B.2.

**Theorem 4.1.** *Consider a tabular setting with finite state-action space $\mathcal{S} \times \mathcal{A}$. Let $Q_t$ and $\widehat{Q}_t$ denote the Q-values at iteration $t$, updated via the standard Bellman optimality equation and the in-distribution bootstrapping variant, respectively. Then, both sequences converge to fixed points $Q^*$ and $\widehat{Q}^*$, i.e., $\lim_{t \to \infty} Q_t = Q^*$ and $\lim_{t \to \infty} \widehat{Q}_t = \widehat{Q}^*$. Furthermore, for all $(s, a) \in \mathcal{S} \times \mathcal{A}$, we have $\widehat{Q}^*(s, a) \leq Q^*(s, a)$, with equality holding if all state-action pairs are visited.*

## 5 Experiments

We design our experiments to evaluate the effectiveness of STAR under various conditions and to validate the impact of our key design components. Specifically, we aim to answer the following questions: $\mathcal{Q}1$: Does STAR achieve strong performance with limited feedback in both online and offline settings? (Sec. 5.3) $\mathcal{Q}2$: Can preference regularization mitigate overfitting in the reward model? (Sec. 5.4) $\mathcal{Q}3$: Does policy regularization alleviate Q-value overestimation compared to prior methods? (Sec. 5.4) $\mathcal{Q}4$: How do the two regularization components complement each other to improve overall learning efficiency? (Sec. 5.4)

Further details about the tasks used in our experiments are provided in Appendix D.

### 5.1 Evaluation Setup

**Tasks.** We evaluate STAR on **18 tasks** drawn from standard benchmarks. In the online setting, we use three locomotion tasks from the DeepMind Control Suite (DMControl) [52] and three robotic manipulation tasks from Meta-World [63, 65]. For the offline setting, we include eight challenging control tasks from D4RL [7] and four robotic manipulation tasks from Robosuite [74]. These tasks span a diverse set of continuous control scenarios, varying in complexity and feedback characteristics.

**Baselines.** We compare STAR against several state-of-the-art methods, covering both online and offline PbRL settings. **Online Methods:** (1) *PEBBLE*[24]: combines unsupervised pretraining with reward relabeling to improve sample efficiency during policy learning. (2) *SURF*[37]: employs temporal data augmentation and pseudo-labels in a semi-supervised framework to enhance policy performance. (3) *MRN* [28]: introduces bi-level optimization for reward learning and represents the current state-of-the-art in online PbRL. **Offline Methods:** (4) *PT*[21]: leverages transformer-based architectures to model non-Markovian rewards and preference-weighting functions, capturing long-term dependencies in offline PbRL. (5) *IPL*[13]: directly optimizes implicit rewards derived from the learned Q-function to align with expert preferences. (6) *DTR*[51]: balances staying close

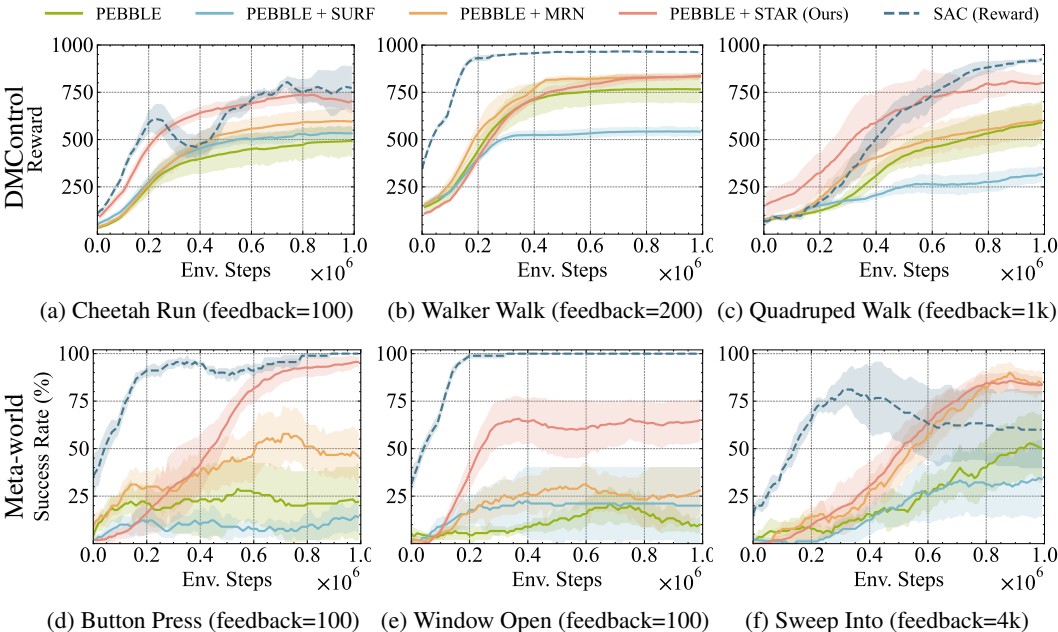

Figure 3: Evaluating curves of all methods on locomotion tasks and robotic manipulation tasks. The solid line presents the mean values, while the shaded area indicates the 78% confidence interval over five runs. The red line is our method.

to the behavior policy with high-return trajectories and choosing actions with high reward labels. **Reward-based Methods:** Since PbRL lacks explicit rewards, we also report results from reward-based algorithms for reference. For online settings, we include (7) *SAC*[11]; for offline settings, we evaluate (8) *IQL*[22] and (9) *TD3+BC* [8]. These comparisons provide a reference for evaluating the performance of STAR relative to methods with access to ground-truth rewards.

**Evaluation Metrics.** We evaluate all algorithms over five independent trials per task and report the mean and standard deviation of their performance. Meta-World tasks (Button Press, Window Open and Sweep Into) are assessed using *success rate*, as they involve goal-conditioned objectives, while DMControl tasks (Cheetah Run, Walker Walk and Quadruped Walk) and offline benchmarks are evaluated based on *ground-truth episode return*, which captures cumulative performance in continuous control settings.

### 5.2 Implementation details

In our experiments, we follow the basic setup employed by prior work [24, 37, 28], which includes unsupervised exploration and an uncertainty-based trajectory sampling strategy. All methods employ an ensemble of three reward models, with outputs bounded in $[-1, 1]$ using a hyperbolic tangent activation (details in Appendix C.1).

To enable systematic and reproducible evaluation, we adopt a scripted teacher, following [24, 37, 28], which generates preference labels by comparing pairs of trajectory segments based on the environment's true reward function. These preferences accurately reflect the underlying rewards, enabling direct quantitative comparison across algorithms. Importantly, agents do not have access to ground-truth rewards and rely solely on preference signals. For offline experiments, we use real human preference data from Kim et al. [21]. The number of preference pairs varies by task complexity: 100 pairs for tasks such as Cheetah Run, Button Press, and Window Open; 500 pairs for tasks like Hopper-medium-replay-v2; and up to 4000 pairs for more complex tasks like Sweep Into. The number of preference labels used per task is detailed in Appendix C.2.

To ensure fair comparisons, all methods are trained with the same network architecture and shared hyperparameters, including learning rate and batch size. The only differences lie in method-specific components such as the loss function. We use publicly available implementations for PEBBLE,

SURF, MRN, and IPL. Additional implementation details, including hyperparameter settings and network architecture, are provided in Appendix C.

## 5.3 Main Results

Our evaluation primarily focuses on training with limited feedback. This refers to an insufficient number of preference labels compared to the feedback required to achieve SAC algorithm performance.

**Online Settings.** We evaluate STAR's performance on three locomotion tasks from DM-Control and three robotic manipulation tasks from Meta-World. As shown in Figure 3, STAR outperforms the baselines in most tasks. In the Cheetah Run task, STAR achieves 96% of the best performance using only 100 preference labels, demonstrating efficient feedback usage. Building on PEBBLE as the backbone,

Table 1: **Summary of normalized scores.** Mean normalized scores (baseline / SAC) across tasks from DMControl and Meta-World.

| Task/Method | PEBBLE | SURF | MRN | STAR (ours) |
|---|---|---|---|---|
| DMControl | 0.704 | 0.550 | 0.781 | 0.881 |
| Meta-world | 0.358 | 0.309 | 0.614 | 1.000 |
| Average | 0.531 | 0.430 | 0.698 | 0.941 (**34.8%** ↑) |

the comparison between the red (STAR) and green (PEBBLE) curves in Figure 3 reveals a significant performance gap. This highlights the effectiveness of our techniques, including preference margin regularization and policy regularization, in enhancing performance with limited preference labels.

Table 2: Averaged normalized scores of all baselines on AntMaze, Gym-Mujoco locomotion tasks, and success rate on Robosuite tasks. All agents training use the same real human preferences dataset from Kim et al. [21]. We train the STAR and IPL, and report the average and standard deviation averaged over 15 episodes. The term 'reward' refers to the use of ground-truth rewards.

| Dataset | IQL (reward) | TD3+BC (reward) | MR | PT | IPL | DTR | STAR (ours) |
|---|---|---|---|---|---|---|---|
| antmaze-medium-play-v2 | 73.88 ± 4.49 | 0.25 ± 0.43 | 31.13 ± 16.96 | 70.13 ± 3.76 | 30.19 ± 4.97 | 67.12 ± 4.22 | 69.0 ± 14.97 |
| antmaze-medium-diverse-v2 | 68.13 ± 10.15 | 0.25 ± 0.43 | 19.38 ± 9.24 | 65.25 ± 3.59 | 24.21 ± 5.12 | 63.23 ± 15.19 | 67.0 ± 17.44 |
| antmaze-large-play-v2 | 48.75 ± 4.35 | 0.0 ± 0.0 | 24.25 ± 14.03 | 42.38 ± 9.98 | 12.46 ± 7.2 | 45.93 ± 14.32 | 50.67 ± 10.2 |
| antmaze-large-diverse-v2 | 44.38 ± 4.47 | 0.0 ± 0.0 | 5.88 ± 6.94 | 19.63 ± 3.70 | 0.0 ± 0.0 | 47.10 ± 12.31 | 48.0 ± 16.0 |
| antmaze-v2 total | 58.79 | 0.13 | 20.16 | 49.35 | 16.72 | 55.85 | **58.67** |
| hopper-medium-replay-v2 | 83.06 ± 15.80 | 64.42 ± 21.52 | 11.56 ± 30.27 | 84.54 ± 4.07 | 73.57 ± 6.7 | 82.12 ± 21.38 | 85.29 ± 5.11 |
| hopper-medium-expert-v2 | 73.55 ± 41.47 | 101.17 ± 9.07 | 57.75 ± 23.70 | 68.96 ± 33.86 | 74.52 ± 0.1 | 91.13 ± 27.20 | 96.75 ± 4.31 |
| walker2d-medium-replay-v2 | 73.11 ± 8.07 | 85.62 ± 4.01 | 72.07 ± 1.96 | 71.27 ± 10.30 | 59.92 ± 5.1 | 73.00 ± 7.13 | 73.91 ± 1.33 |
| walker2d-medium-expert-v2 | 107.75 ± 2.02 | 110.03 ± 0.36 | 108.32 ± 3.87 | 110.13 ± 0.21 | 108.51 ± 0.6 | 108.39 ± 5.28 | 110.12 ± 0.24 |
| locomotion-v2 total | 84.37 | 90.31 | 62.43 | 83.72 | 79.13 | 88.66 | **91.52** |
| lift-ph | 96.75 ± 1.83 | - | 84.75 ± 6.23 | 91.75 ± 5.90 | 97.6 ± 2.9 | 94.51 ± 3.67 | 98.0 ± 4.0 |
| lift-mh | 86.75 ± 2.82 | - | 91.00 ± 4.00 | 86.75 ± 5.95 | 87.2 ± 5.3 | 92.69 ± 4.81 | 93.0 ± 8.94 |
| can-ph | 74.50 ± 6.82 | - | 68.00 ± 9.13 | 69.67 ± 5.89 | 74.8 ± 2.4 | 67.12 ± 9.76 | 70.0 ± 13.56 |
| can-mh | 56.25 ± 8.78 | - | 47.50 ± 3.51 | 50.50 ± 6.48 | 57.6 ± 5.0 | 42.71 ± 9.78 | 47.0 ± 9.8 |
| robosuite total | 78.56 | - | 72.81 | 74.66 | **79.3** | 74.26 | 77.0 |

To further illustrate the results, we summarize the mean normalized scores (baseline score / SAC score) across tasks from DMControl and Meta-World in Table 1. It shows that STAR surpasses the state-of-the-art (MRN) by 34.8%, indicating STAR significantly improves the feedback efficiency of PbRL methods across a range of complex tasks.

**Offline Settings.** We benchmark STAR against several offline PbRL algorithms using the D4RL [7] and Robosuite [74] robotics datasets, incorporating real-human preference data from Kim et al. [21]. To avoid relying on out-of-sample actions, we extract the policy using advantage weighted regression [38, 53, 34]. Our evaluation includes comparisons with two state-of-the-art offline PbRL algorithms, IPL and PT, as well as two significant offline RL algorithms. Since TD3 serves as our foundational RL algorithm, we also present results from TD3 enhanced with Behavior Cloning (TD3+BC) [8].

Additionally, we compare different reward model architectures: MR (MLP-based), LSTM (LSTM-based), and PT (Transformer-based), corresponding to models proposed by Christiano et al. [4], Early et al. [6], and Kim et al. [21], respectively. The results in Table 2 show that STAR outperforms all baselines on most tasks. On D4RL, our method achieves an average performance improvement of 29.7% compared to state-of-the-art methods. Notably, it exhibits a clear advantage in challenging tasks such as antmaze-large, where it nearly matches the performance of IQL with task rewards and

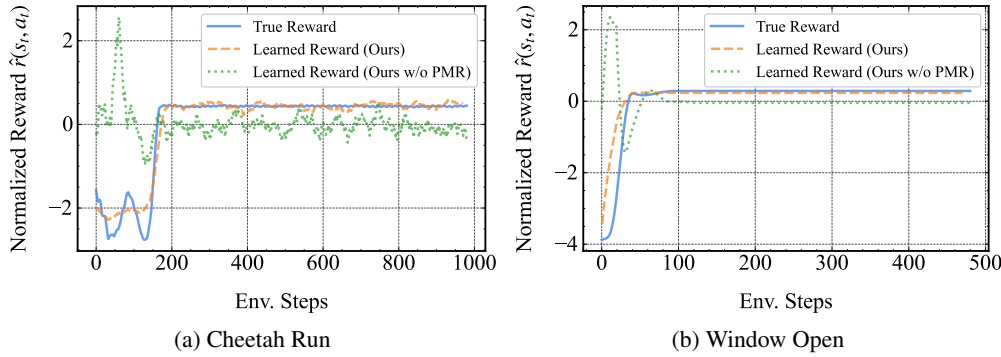

(a) Cheetah Run            (b) Window Open

Figure 4: **Quality of Learned Reward.** Time series plots of the normalized learned reward (dashed line) and the ground-truth reward (solid line) obtained from rollouts of a policy trained with STAR.

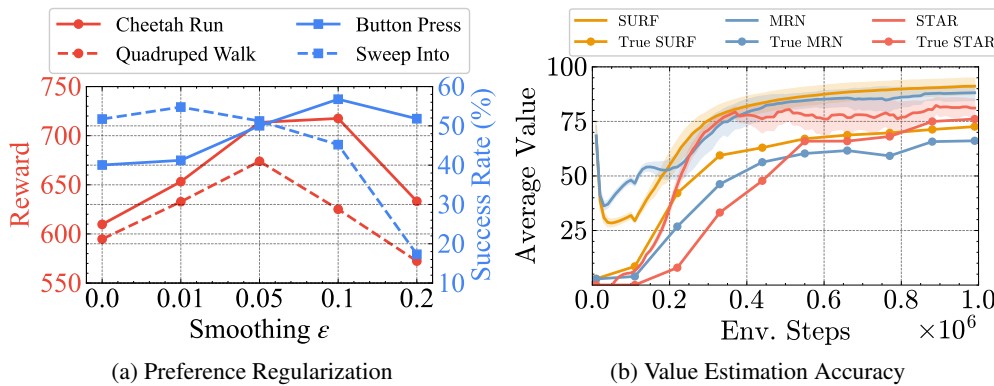

(a) Preference Regularization          (b) Value Estimation Accuracy

Figure 5: Ablation studies evaluating (a) the impact of the parameter $\varepsilon$ in preference margin regularization, and (b) the overestimation bias in value estimates.

even surpasses IQL in certain cases. These results highlight the effectiveness of our approach across diverse tasks in offline settings.

### 5.4 Ablation Studies

**Quality of Learned Reward.** To evaluate the impact of preference regularization on reward model quality, we compare the learned reward model with the ground-truth reward function on Cheetah Run and Window Open tasks. As shown in Figure 4, we plot time series curves for our method (STAR), its ablated variant without PMR, and the ground-truth reward. The results demonstrate that STAR produces a reward function that more closely tracks the ground-truth reward across time steps. In contrast, the variant without PMR exhibits higher variance and deviates significantly from the true reward, indicating that PMR contributes to improved reward model calibration.

**Accuracy of Value Estimation.** We assess the accuracy of value estimation in STAR by tracking the value estimate trajectory during the learning process on Cheetah Run. Figure 5b charts the average value estimate over 10,000 states and compares it to an estimate of the true value. The true value is computed as the average discounted return obtained by following the current policy using the ground truth reward. A clear overestimation bias is observed during learning, with SURF overestimating by 32% and MRN by 25%. When constrained by the policy regularizer, STAR significantly reduces overestimation bias to just 7%, leading to a more accurate Q-function. This reduction in bias directly improves value estimation, enabling a more stable and effective learning process in PbRL. Quantitatively, the overestimation error decreases by 72%, resulting in a more reliable learning trajectory and contributing to better policy performance across a variety of tasks.

**Contribution of each technique.** To assess the individual contributions of each technique and their interaction effects, we conduct an ablation study on preference margin regularization (PMR) and

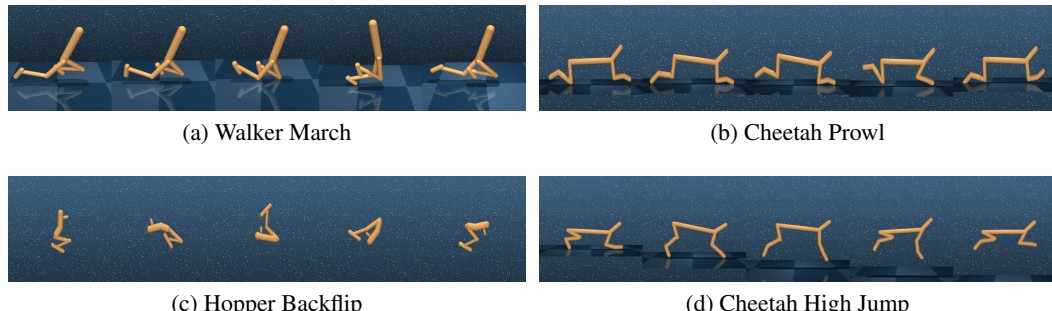

| (a) Walker March | (b) Cheetah Prowl |
|---|---|

| (c) Hopper Backflip | (d) Cheetah High Jump |
|---|---|

Figure 7: **Human Experiments.** Novel skills learned from human feedback.

policy regularization (PR). Table 3 reports the performance on three tasks: Cheetah Run, Window Open, and Sweep Into. Removing either PMR or PR leads to a notable performance drop across all tasks, with the absence of both causing the most severe degradation. These results highlight that both techniques are essential and complementary in improving the robustness and effectiveness of STAR.

We further investigate the impact of the parameter $\varepsilon$ in PMR. Figure 5a shows how varying $\varepsilon$ influences the performance of STAR. Interestingly, even a small value (e.g., $\varepsilon = 0.01$) leads to noticeable performance gains, indicating the sensitivity of the reward model to minor regularization. However, excessively large $\varepsilon$ values introduce estimation bias, consistent with observations in prior work [12, 33]. In addition, we observe that tasks with more feedback (e.g., Quadruped Walk, Sweep Into) tend to perform best with smaller $\varepsilon$ values, whereas tasks with fewer feedback (e.g., Cheetah Run, Button Press) benefit from larger $\varepsilon$. This suggests that the severity of reward overfitting diminishes with increased feedback, reducing the need for aggressive smoothing.

Table 3: **Contribution of Each Technique.** Performance on Cheetah Run is measured by episode return, while Window Open and Sweep Into are measured by success rate.

| Task | Ours | Ours w/o PMR | Ours w/o PR | Ours w/o both |
|---|---|---|---|---|
| Cheetah Run | 713.5 | 609.2 | 579.7 | 491.5 |
| Window Open | 65.0 | 50.0 | 41.2 | 10.0 |
| Sweep Into | 81.2 | 59.7 | 51.4 | 41.6 |

**Effect of Feedback Quantity.** To further investigate how the number of feedback signals influences performance, we evaluate all methods on Walker Walk with varying amounts of feedback $N \in \{100, 200, 500, 1000\}$. As shown in Figure 6, performance improves across all methods as feedback increases, likely due to a more accurate reward model. Notably, STAR consistently outperforms all baselines, even under limited feedback, highlighting its ability to learn a more reliable Q-function and maintain strong performance with fewer preference labels.

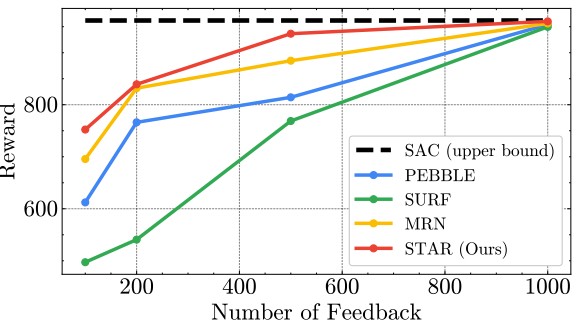

Figure 6: **Effect of Feedback Quantity on Walker.**

**Human Experiments.** We demonstrate that agents trained with STAR using human feedback can acquire novel and diverse skills, as illustrated in Figure 7. Specifically, we showcase: (a) a Walker lowering its center of gravity and marching, (b) a Cheetah agent prowling, (c) a Hopper performing a backflip, and (d) a Cheetah executing a high jump—each trained with only 50 human preference queries. These results underscore the effectiveness of human feedback in guiding complex behavior acquisition via STAR, particularly in settings where manual reward design is non-trivial. Videos of all behaviors are included in the supplementary material.

# 6 Conclusion

In this work, we introduce STAR, a novel preference-based RL algorithm that significantly enhances feedback efficiency. By integrating preference margin regularization and policy regularization, STAR outperforms prior methods and achieves strong performance across a range of complex tasks in both online and offline settings. A key strength of our method lies in its ability to learn effectively from a limited number of preference labels. Our analysis reveals two main factors driving this efficiency: **(1)** mitigating overestimation bias through conservative policy regularization, leading to more accurate Q-function estimates; and **(2)** reducing reward model overfitting via preference margin regularization. Additionally, human experiments demonstrate that STAR enables agents to acquire diverse and novel skills, reinforcing its practical potential. By improving the efficiency of PbRL, STAR takes a step toward making preference-driven learning more applicable to real-world robotic systems.

# 7 Limitation

Despite the significant improvement in feedback efficiency achieved by STAR, this work primarily focuses on state-based inputs. Extending the method to handle partially observable or high-dimensional inputs, such as raw RGB observations, remains an important direction for future research. Moreover, all experiments are conducted in simulation. A key next step for future is to investigate the applicability of STAR to real-world RL, enabling robots to acquire manipulation skills directly on physical platforms.

## Acknowledgments

This work is partially supported by National Key R&D Program of China (2024YFC3505402) and Project No. 20240313, Zhongguancun Academy, Beijing, China (100094). We sincerely thank the anonymous reviewers for their helpful comments and suggestions, which improved this paper.

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

# Supplementary Material

## Table of Contents

# A Full Procedure of STAR

## A.1 Overview

Our method builds on the PbRL framework PEBBLE [24], which serves as a backbone for many online PbRL algorithms [37, 28]. In this paper, we propose a generic and flexible algorithm that can be seamlessly integrated with any PbRL algorithm and adapted to various settings, including discrete and continuous action spaces, as well as online and offline environments. The detailed procedures of our method are presented in Algorithm 1 for online settings and Algorithm 2 for offline settings.

For online settings, STAR introduces two distinct constrained operators for conservative Q-value estimation: $\max$ for discrete action spaces and $\mathcal{T}$ for continuous action spaces. These operators ensure more reliable policy learning by constraining overestimation in different action domains. In offline settings, our method first trains the reward model $\widehat{r}_\psi$ using pre-collected human preference data, then proceeds to policy learning, carefully addressing the challenges of limited feedback and bias inherent in offline RL.

---

**Algorithm 1** STAR (Online)

---

**Require:** preference query frequency $K$, number of human's preference labels per session $M$
1: Initialize parameters of $Q_\theta, \pi_\phi, \widehat{r}_\psi, Q_\xi$ and preference dataset $\mathcal{D} \leftarrow \emptyset$
2: Initialize replay buffer $\mathcal{B}$ and $\pi_\theta$ with unsupervised exploration
3: **for** each iteration **do**
4:      Take action $a_t \sim \pi_\theta$ and collect $s_{t+1}$
5:      **if** iteration % $K == 0$ **then**
6:         // Query preference
7:         Sample pair of trajectories $(\sigma^0, \sigma^1)$ and query human for $y$
8:         Store preference data into dataset $\mathcal{D} \leftarrow \mathcal{D} \cup \{(\sigma^0, \sigma^1, y)\}$
9:         // Reward learning
10:        Sample batch $\{(\sigma^0, \sigma^1, y)_i\}_{i=1}^n$ from $\mathcal{D}$
11:        Optimize Equation (2) to update $\widehat{r}_\psi$
12:        Relabel the replay buffer $\mathcal{B}$ using $\widehat{r}_\psi$
13:      **end if**
14:      // Estimate conservative $\widehat{Q}$
15:      Store transition $(s_t, a_t, \widehat{r}_\psi(s_t, a_t), s_{t+1})$ into replay buffer $\mathcal{B}$
16:      Drive $\widehat{Q}$ via Equation (11) (discrete setting)
17:      Update $Q_\xi$ via Equation (9) (continuous setting)
18:      // Policy regularization
19:      Update $Q_\theta$ according to Equation (12).(discrete setting)
20:      Update $Q_\theta$ and $\pi_\phi$ according to Equation (4) and Equation (3), respectively.(continuous setting)
21: **end for**
**Ensure:** policy $\pi_\phi$

---

---

**Algorithm 2** STAR (Offline)

---

**Require:** preference dataset $\mathcal{D}$, dataset $\mathcal{B}$
1: Initialize parameters of $Q_\theta, \pi_\phi, \widehat{r}_\psi$
2: // Reward learning
3: Optimize Equation (2) to update $\widehat{r}_\psi$ with preference data from $\mathcal{D}$
4: Label the dataset $\mathcal{B}$ via $\widehat{r}_\psi$
5: **for** each iteration **do**
6:      Sample a batch $(s, a, \widehat{r}_\psi(s, a), s')$ from $\mathcal{B}$
7:      // Estimate conservative $\widehat{Q}$
8:      Drive $\widehat{Q}$ via Equation (11) (discrete setting)
9:      Update $Q_\xi$ via Equation (9) (continuous setting)
10:      // Policy regularization
11:      Update $Q_\theta$ according to Equation (12).(discrete setting)
12:      Update $Q_\theta$ and $\pi_\phi$ according to Equation (4) and Equation (3), respectively.(continuous setting)
13: **end for**
**Ensure:** policy $\pi_\phi$

---

## A.2 Discrete Settings

In online settings with discrete action spaces, we provide an efficient implementation for bootstrapping a conservative estimate $\widehat{Q}$. By leveraging well-supported state-action pairs from the current replay memory, we generate conservative Q-value estimates that mitigate overestimation risks. Specifically, we structure the replay memory as a graph for discrete action spaces, following a similar approach to Zhu et al. [72], Hong et al. [15], Zhang et al. [64]. This graph-based structure allows us to efficiently perform conservative estimations by identifying connections between state-action pairs, enabling rapid updates without incurring significant computational overhead.

**Conservative Estimate $\widehat{Q}$.** We structure the replay memory as a dynamic and directed graph, denoted as $\mathcal{G} = (\mathcal{V}, \mathcal{E})$. Each vertex in this graph represents a state $s$ paired with its associated action value estimation $\widehat{Q}(s, \cdot)$, forming the vertex set: $\mathcal{V} = \{s | (s, \widehat{Q}(s, \cdot))\}$. Each directed edge represents a transition from state $s$ to state $s'$ via action $a$, storing both the estimated reward $\widehat{r}_\psi(s, a)$ and the transition count $N(s, a, s')$. These elements are essential for updating the model based on experience. The edge set is represented as $\mathcal{E} = \{s \xrightarrow{a} s' | (a, \widehat{r}_\psi(s, a), N(s, a, s'), \widehat{Q}(s, a))\}$. To achieve efficient querying, each vertex and edge is assigned a unique key through a hash function, ensuring a query time complexity of $\mathcal{O}(1)$. This allows the graph to handle dynamic updates without significant overhead. Each vertex $v$ contains an action set $\partial\mathcal{A}(s)$, which tracks the actions executed in state $s$, facilitating in-sample updates and improving learning efficiency. Similar to conventional replay memory, this graph stores the most recent experiences while maintaining a fixed memory size to prevent excessive memory usage.

This graph updating contains two primary components: estimation updating and reward relabeling. When a new transition $(s, a, \widehat{r}_\psi(s, a), s')$ is observed, a new vertex and edge are added to the graph based on the previously outlined structure, initializing $\widehat{Q}(s, a) = 0$ and setting $N(s, a, s') = 1$. If the edge for the transition already exists, the visit count is incremented: $N(s, a, s') \leftarrow N(s, a, s') + 1$, allowing the graph to adapt to repeated experiences and refine its estimation over time.

For updating the action value estimate $\widehat{Q}$, we sample a subset of graph vertices $\partial\mathcal{V} \subseteq \mathcal{V}$ in reverse order, similar to the techniques used by Rotinov [41], Lee et al. [25]. This reverse-order sampling improves the efficiency of the updates by leveraging the structure of the graph, allowing for rapid convergence of $\widehat{Q}$. During this process, the max-operator is constrained to operate over the in-sample action set $\partial\mathcal{A}(s)$ rather than the entire action space. This ensures that the method remains focused on well-explored actions, reducing the risk of overestimation by avoiding out-of-sample actions.

The value iteration process is defined as:

$$\widehat{Q}(s, a) \leftarrow \sum_{s' \in \mathcal{S}} \widehat{p}(s'|s, a) \Big[ \widehat{r}_\psi(s, a) + \gamma \max_{a' \in \partial\mathcal{A}(s')} \widehat{Q}(s', a') \Big], \tag{11}$$

where the empirical transition dynamics $\widehat{p}(s'|s, a) = N(s, a, s') / \sum_{s'} N(s, a, s')$ are computed from the graph. This update rule ensures that unseen actions are not queried during value estimation, thereby preventing overestimation.

For reward relabeling, each time the reward model $\widehat{r}_\psi$ is updated, all past experiences in the graph are relabeled according to the updated model. This maximizes the utility of historical data and accounts for non-stationary reward functions. While this relabeling process can be computationally intensive as the number of stored transitions grows, it allows the algorithm to adapt to changing reward landscapes and ensures that the learning process remains up-to-date with the most accurate reward information.

**Updating $Q_\theta$.** In scenarios with discrete actions, we define $\widehat{\pi}$ as the Boltzmann policy derived from $\widehat{Q}$, where $\widehat{\pi}(s) = \text{Softmax}_{a \in \partial\mathcal{A}(s)}(\widehat{Q}(s, \cdot))$. This policy is inherently conservative, considering only the support set $\partial\mathcal{A}(s)$ for a given state $s$. Similarly, the policy $\pi$, derived from the $Q_\theta$ network, is expressed as $\pi(s) = \text{Softmax}_{a \in \partial\mathcal{A}(s)}(Q_\theta(s, \cdot))$. Total loss is defined as follows:

$$\mathcal{L}_{\text{discrete}}(\theta) = \mathbb{E}_{\tau_t \sim \mathcal{G}} \Big[ (Q_\theta(s, a) - y)^2 \Big] + \eta \mathcal{L}_{\text{reg}}(\theta), \tag{12}$$

where $y = \widehat{r}_\psi(s, a) + \gamma \max_{a'} Q(s', a')$ is the Q target, $\tau_t = (s_t, a_t, s_{t+1}, \widehat{r}_\psi(s_t, a_t))$ is the transition and $\eta$ is the weight factor.

## B  Theoretical Analysis

### B.1  Reward Overfitting Analysis

**Why Reward Overfitting?** For reward learning, the logits output by the reward model are transformed into probabilities using a softmax function, ensuring that the resulting probabilities sum to 1. The loss function in equation (13) is used to optimize $r_\psi$ as follows:

$$\mathcal{L}_{\text{reward}}(\psi) = - \mathop{\mathbb{E}}_{(\sigma^0,\sigma^1,y)\sim\mathcal{D}} \left[ y(0) \log P_\psi[\sigma^0 \succ \sigma^1] + y(1) \log P_\psi[\sigma^1 \succ \sigma^0] \right]. \tag{13}$$

In the optimal scenario, we aim for the predicted probability of the preferred segment to be 1, while the rejected segment's predicted probability should be 0. This can be represented mathematically as:

$$\frac{\exp(\sum_t \widehat{r}_\psi(s_t^0, a_t^0))}{\exp(\sum_t \widehat{r}_\psi(s_t^0, a_t^0)) + \exp(\sum_t \widehat{r}_\psi(s_t^1, a_t^1))} = 1$$
$$\frac{\exp(\sum_t \widehat{r}_\psi(s_t^1, a_t^1))}{\exp(\sum_t \widehat{r}_\psi(s_t^0, a_t^0)) + \exp(\sum_t \widehat{r}_\psi(s_t^1, a_t^1))} = 0 \tag{14}$$

Expanding the softmax expressions in equation (14) yields the following analytical solution:

$$\exp(\sum_t \widehat{r}_\psi(s_t^0, a_t^0)) = \exp(\sum_t \widehat{r}_\psi(s_t^0, a_t^0)) + \exp(\sum_t \widehat{r}_\psi(s_t^1, a_t^1))$$
$$\sum_t \widehat{r}_\psi(s_t^0, a_t^0) = \log\left( \exp(\sum_t \widehat{r}_\psi(s_t^0, a_t^0)) + \exp(\sum_t \widehat{r}_\psi(s_t^1, a_t^1)) \right)$$
$$\exp(\sum_t \widehat{r}_\psi(s_t^1, a_t^1)) = 0$$
$$\sum_t \widehat{r}_\psi(s_t^1, a_t^1) = -\infty \tag{15}$$

Upon examining this analytical solution in equation (15), we observe that for the reward model to predict optimal behavior, the preferred segments must be assigned a constant value, while the rejected segments approach negative infinity. However, achieving this theoretical optimal solution is practically impossible, as it would lead to overfitting. In practice, this results in the reward model being overly confident in its predictions, causing poor generalization to new data.

**Why preference margin regularization effect?** To address this issue, we introduce a margin technique inspired by Szegedy et al. [48]. By introducing a bounded objective in reward optimization, we constrain the reward model, preventing it from overoptimizing the loss function and ensuring that the analytical solution remains bounded. The specific update rule is as follows:

$$y(i,j) = \begin{cases} (1-\varepsilon, \varepsilon), & \text{if } \sigma^i \succ \sigma^j \\ (0.5, 0.5), & \text{otherwise.} \end{cases} \tag{16}$$

where $\varepsilon$ is margin factor.

Under preference margin regularization, we aim for the predicted probability of the preferred segment to be $1-\varepsilon$, while the rejected segment's predicted probability should be $\varepsilon$. This can be represented mathematically as:

$$\frac{\exp(\sum_t \widehat{r}_\psi(s_t^0, a_t^0))}{\exp(\sum_t \widehat{r}_\psi(s_t^0, a_t^0)) + \exp(\sum_t \widehat{r}_\psi(s_t^1, a_t^1))} = 1 - \varepsilon$$
$$\frac{\exp(\sum_t \widehat{r}_\psi(s_t^0, a_t^1))}{\exp(\sum_t \widehat{r}_\psi(s_t^0, a_t^0)) + \exp(\sum_t \widehat{r}_\psi(s_t^1, a_t^1))} = \varepsilon \tag{17}$$

Expanding the softmax expressions in equation (17) yields the following analytical solution:

$$\exp(\sum_t \widehat{r}_\psi(s_t^0, a_t^0)) = (1 - \varepsilon) * \exp(\sum_t \widehat{r}_\psi(s_t^0, a_t^0)) + (1 - \varepsilon) * \exp(\sum_t \widehat{r}_\psi(s_t^1, a_t^1))$$

$$\varepsilon * \exp(\sum_t \widehat{r}_\psi(s_t^0, a_t^0)) = (1 - \varepsilon) * \exp(\sum_t \widehat{r}_\psi(s_t^1, a_t^1))$$

$$\sum_t \widehat{r}_\psi(s_t^0, a_t^0) - \sum_t \widehat{r}_\psi(s_t^1, a_t^1) = \log\left(\frac{1 - \varepsilon}{\varepsilon}\right)$$

(18)

From the derivation in equation (18), we observe that after applying preference margin regularization, the objective shifts from maximizing the difference between preferred and rejected samples to maintaining a controlled margin. Specifically, preference margin regularization introduces a bounded margin of $\log\left(\frac{1-\varepsilon}{\varepsilon}\right)$ between positive and negative samples. This ensures that we do not overly optimize the loss during reward learning, which could lead to overfitting.

The key insight behind preference margin regularization is that it limits the optimization process, preventing the model from becoming overly confident in its predictions by capping the margin between preferred and rejected samples. By keeping the margin bounded, preference margin regularization helps stabilize the learning process, reducing the risk of overfitting and improving the model's ability to generalize to unseen states.

### B.2 Proof of Theorem 4.1

In this section, we analyze the property of $\widehat{Q}$ in finite state-action space $\mathcal{S} \times \mathcal{A}$. The proof of $\lim_{t\to\infty} Q_t = Q^*$ has been well-established in previous work [40, 18, 31]. Then the proof of $\lim_{t\to\infty} \widehat{Q}_t = \widehat{Q}^*$ is similar. We first prove the empirical Bellman operator Equation (21) is a $\gamma$-contraction operator under the supremum norm. Then when updating in a sampling manner as Equation (22), it can be considered as a random process. Borrowing an auxiliary result from stochastic approximation, we prove it satisfies the conditions that guarantee convergence. Finally, to prove $\widehat{Q}^*$ lower-bounds $Q^*$, we rewrite $\widehat{Q}^*(s, a) - Q^*(s, a)$ based on the standard and empirical Bellman operators. When the data covers the whole state-action space, we naturally have $\widehat{Q}^* = Q^*$.

For proof simplicity, we use $\beta$ denotes policies that interact with the environment and form the current replay memory. We first show existing results for Bellman learning in Equation (20), and then prove Theorem 4.1 in three steps. The Bellman (optimality) operator $\mathcal{B}$ is defined as:

$$(\mathcal{B}Q)(s, a) = \sum_{s'\in\mathcal{S}} P(s'|s, a)[r + \gamma \max_{a'} Q(s', a')]. \tag{19}$$

Previous works have shown the operator $\mathcal{B}$ is a $\gamma$-contraction with respect to supremum norm:

$$\|\mathcal{B}Q_1 - \mathcal{B}Q_2\|_\infty \le \gamma\|Q_1 - Q_2\|_\infty,$$

the supremum norm $\|v\|_\infty = \max_{1\le i\le d} |v_i|$, $d$ is the dimension of vector $v$. Following Banach's fixed-point theorem, $Q$ converges to optimal action value $Q^*$ if we consecutively apply operator $\mathcal{B}$ to $Q$, $\lim_{n\to\infty}(\mathcal{B})^n Q = Q^*$.

Further, the update rule in Equation (20), i.e. $Q$-learning, is a sampling version that applies the $\gamma$-contraction operator $\mathcal{B}$ to $Q$.

$$Q(s, a) \leftarrow r(s, a) + \gamma \max_{a'} Q(s', a'). \tag{20}$$

It can be considered as a random process and will converge to $Q^*$, $\lim_{t\to\infty} Q_t = Q^*$, with some mild conditions [49, 40, 18, 31].

Similarly, we define the empirical Bellman (optimality) operator $\hat{\mathcal{B}}$ as:

$$(\hat{\mathcal{B}}\widehat{Q})(s, a) = \sum_{s'\in\mathcal{S}} P(s'|s, a)[r + \gamma \max_{a':\beta(a'|s')>0} \widehat{Q}(s', a')]. \tag{21}$$

And the sampling version is:

$$\widehat{Q}(s, a) \leftarrow r + \gamma \max_{a':\beta(a'|s')>0} \widehat{Q}(s', a'), \tag{22}$$

We split Theorem 4.1 into three lemmas. We first show $\hat{\mathcal{B}}$ is a $\gamma$-contraction operator under supremum norm, thus converges to optimal action value $\widehat{Q}^*$, $\lim_{n\to\infty}(\mathcal{B})^n\widehat{Q} = \widehat{Q}^*$. Then we show the sampling-based update rule in Equation (22) converges to $\widehat{Q}^*$, $\lim_{t\to\infty}\widehat{Q}_t = \widehat{Q}^*$. Finally, we show $\widehat{Q}^*$ lower-bounds $Q^*$, $\widehat{Q}^*(s,a) - Q^*(s,a) \leq 0, \forall(s,a) \in \mathcal{S}\times\mathcal{A}$. And when the data covers the whole state-action space, i.e. $\beta(a|s) > 0$ for all state-action pairs, we naturally have $\widehat{Q}^*(s,a) = Q^*(s,a)$.

**Lemma B.1.** *The operator $\hat{\mathcal{B}}$ defined in Equation* (21) *is a $\gamma$-contraction operator under supremum norm,*
$$\|\hat{\mathcal{B}}\widehat{Q}_1 - \hat{\mathcal{B}}\widehat{Q}_2\|_\infty \leq \gamma\|\widehat{Q}_1 - \widehat{Q}_2\|_\infty.$$

*Proof.* We can rewrite $\|\hat{\mathcal{B}}\widehat{Q}_1 - \hat{\mathcal{B}}\widehat{Q}_2\|_\infty$ as

$$\|\hat{\mathcal{B}}\widehat{Q}_1 - \hat{\mathcal{B}}\widehat{Q}_2\|_\infty$$
$$= \max_{s,a}\Big|\sum_{s'\in\mathcal{S}} P(s'|s,a)[r + \gamma\max_{a'_1:\beta(a'_1|s')>0}\widehat{Q}_1(s',a'_1)] - P(s'|s,a)[r + \gamma\max_{a'_2:\beta(a'_2|s')>0}\widehat{Q}_2(s',a'_2)]\Big|$$
$$= \max_{s,a}\gamma\Big|\sum_{s'\in\mathcal{S}} P(s'|s,a)[\max_{a'_1:\beta(a'_1|s')>0}\widehat{Q}_1(s',a'_1) - \max_{a'_2:\beta(a'_2|s')>0}\widehat{Q}_2(s',a'_2)]\Big|$$
$$\leq \max_{s,a}\gamma\sum_{s'\in\mathcal{S}} P(s'|s,a)\Big|\max_{a'_1:\beta(a'_1|s')>0}\widehat{Q}_1(s',a'_1) - \max_{a'_2:\beta(a'_2|s')>0}\widehat{Q}_2(s',a'_2)\Big|$$
$$\leq \max_{s,a}\gamma\sum_{s'\in\mathcal{S}} P(s'|s,a)\max_{\tilde{a}:\beta(\tilde{a}|s')>0}\Big|\widehat{Q}_1(s',\tilde{a}) - \widehat{Q}_2(s',\tilde{a})\Big|$$
$$\leq \max_{s,a}\gamma\sum_{s'\in\mathcal{S}} P(s'|s,a)\max_{\tilde{s},\tilde{a}:\beta(\tilde{a}|\tilde{s})>0}\Big|\widehat{Q}_1(\tilde{s},\tilde{a}) - \widehat{Q}_2(\tilde{s},\tilde{a})\Big|$$
$$= \max_{s,a}\gamma\sum_{s'\in\mathcal{S}} P(s'|s,a)\|\widehat{Q}_1 - \widehat{Q}_2\|_\infty$$
$$= \gamma\|\widehat{Q}_1 - \widehat{Q}_2\|_\infty,$$

where the last line follows from $\sum_{s'\in\mathcal{S}} P(s'|s,a) = 1$. $\qquad\square$

To show the sampling-based update rule in Equation (22) converges to $\widehat{Q}^*$, we borrow an auxiliary result from stochastic approximation [40, 18].

**Theorem B.2.** *The random process $\{\Delta_t\}$ taking values in $\mathbb{R}^n$ and defined as*
$$\Delta_{t+1}(x) = (1 - \alpha_t(x))\Delta_t(x) + \alpha_t(x)F_t(x) \tag{23}$$
*converges to zero w.p.1 under the following assumptions:*

*(1) $0 \leq \alpha_t \leq 1$, $\sum_t \alpha_t(x) = \infty$ and $\sum_t \alpha_t^2(x) < \infty$;*

*(2) $\|\mathbb{E}[F_t(x)|\mathcal{F}_t]\|_W \leq \gamma\|\Delta_t\|_W$, with $\gamma < 1$;*

*(3) $Var[F_t(x)|\mathcal{F}_t] \leq C(1 + \|\Delta_t\|_W^2)$, for $C > 0$.*

$W$ is a norm. In our proof it is a supremum norm.

*Proof.* See Robbins and Monro [40], Jaakkola et al. [18]. $\qquad\square$

**Lemma B.3.** *Given any initial estimation $\widehat{Q}_0$, the following update rule:*
$$\widehat{Q}_{t+1}(s_t,a_t) = \widehat{Q}_t(s_t,a_t) + \alpha_t(x_t,a_t)[r_t + \gamma\max_{a:\beta(a|s_{t+1})>0}\widehat{Q}_t(s_{t+1},a) - \widehat{Q}_t(s_t,a_t)], \tag{24}$$
*converges w.p.1 to the optimal action-value function $\widehat{Q}^*$ if*
$$0 \leq \alpha_t(s,a) \leq 1, \quad \sum_t \alpha_t(s,a) = \infty \quad and \quad \sum_t \alpha_t^2(s,a) < \infty,$$
*for all $(s,a) \in \mathcal{S}\times\mathcal{A}$.*

*Proof.* Based on Theorem B.2, we prove the update rule in Equation (24) converges.

Rewrite Equation (24) as

$$\widehat{Q}_{t+1}(s_t, a_t) = (1 - \alpha_t(s_t, a_t))\widehat{Q}_t(s_t, a_t) + \alpha_t(x_t, a_t)[r_t + \gamma \max_{a:\beta(a|s_{t+1})>0} \widehat{Q}_t(s_{t+1}, a)]$$

Subtract $\widehat{Q}^*(s_t, a_t)$ from both sides:

$$\widehat{Q}_{t+1}(s_t, a_t) - \widehat{Q}^*(s_t, a_t)$$
$$= (1 - \alpha_t(s_t, a_t))(\widehat{Q}_t(s_t, a_t) - \widehat{Q}^*(s_t, a_t)) + \alpha_t(x_t, a_t)[r_t + \gamma \max_{a:\beta(a|s_{t+1})>0} \widehat{Q}_t(s_{t+1}, a) - \widehat{Q}^*(s_t, a_t)]$$

Let

$$\Delta_t(s, a) = \widehat{Q}(s, a) - \widehat{Q}^*(s, a) \tag{25}$$

and

$$F_t(s, a) = r + \gamma \max_{a':\beta(a'|s')>0} \widehat{Q}_t(s', a') - \widehat{Q}^*(s, a). \tag{26}$$

We get the same random process shown in Theorem B.2 Equation (23). Then, proving $\lim_{t\to\infty} \widehat{Q}_t = \widehat{Q}^*$ is the same as proving $\Delta_t(s, a)$ converges to zero with probability 1. We only need to show the assumptions in Theorem B.2 are satisfied under the definitions of Equations (25) and (26).

Theorem B.2 (1) is the same as the condition in Lemma B.3. It is easy to achieve, for example, we can choose $\alpha_t(s, a) = 1/t$.

For Theorem B.2 (2), we have

$$\mathbb{E}[F_t(s, a)|\mathcal{F}_t] = \sum_{s'\in\mathcal{S}} P(s'|s, a)[r + \gamma \max_{a':\beta(a'|s')} \widehat{Q}_t(s', a') - \widehat{Q}^*(s, a)]$$
$$= (\hat{\mathcal{B}}\widehat{Q}_t)(s, a) - \widehat{Q}^*(s, a)$$
$$= (\hat{\mathcal{B}}\widehat{Q}_t)(s, a) - (\hat{\mathcal{B}}\widehat{Q}^*)(s, a)$$

Thus,

$$\|\mathbb{E}[F_t(s, a)|\mathcal{F}_t]\|_\infty = \|(\hat{\mathcal{B}}\widehat{Q}_t) - (\hat{\mathcal{B}}\widehat{Q}^*)\|_\infty$$
$$\leq \gamma\|\widehat{Q}_t - \widehat{Q}^*\|_\infty$$
$$= \gamma\|\Delta_t\|_\infty,$$

with $\gamma < 1$.

For Theorem B.2 (3), we have

$$Var[F_t(s)|\mathcal{F}_t] = \mathbb{E}[F_t(s) - \mathbb{E}[F_t(s)|\mathcal{F}_t]|\mathcal{F}_t]^2$$
$$= \mathbb{E}[F_t(s) - ((\hat{\mathcal{B}}\widehat{Q}_t)(s, a) - (\hat{\mathcal{B}}\widehat{Q}^*)(s, a))]^2$$
$$= \mathbb{E}[r + \gamma \max_{a':\beta(a'|s')>0} \widehat{Q}_t(s', a') - \widehat{Q}^*(s, a) - ((\hat{\mathcal{B}}\widehat{Q}_t)(s, a) - (\hat{\mathcal{B}}\widehat{Q}^*)(s, a))]^2$$
$$= \mathbb{E}[r + \gamma \max_{a':\beta(a'|s')>0} \widehat{Q}_t(s', a') - (\hat{\mathcal{B}}\widehat{Q}_t)(s, a)]^2$$
$$= Var[r + \gamma \max_{a':\beta(a'|s')>0} \widehat{Q}_t(s', a')|\mathcal{F}_t]$$

We add and minus a $\widehat{Q}^*$ term to make it close to the RHS in Theorem B.2 (3):

$$Var[r + \gamma \max_{a':\beta(a'|s')>0} \widehat{Q}^*(s', a') + \gamma \max_{a':\beta(a'|s')>0} \widehat{Q}_t(s', a') - \gamma \max_{a':\beta(a'|s')>0} \widehat{Q}^*(s', a')|\mathcal{F}_t]$$

Since $r$ is bounded, thus $r + \gamma \max_{a':\beta(a'|s')>0} \widehat{Q}^*(s', a')$ is bounded. And clearly the second part $\max_{a':\beta(a'|s')>0} \widehat{Q}_t(s', a') - \max_{a':\beta(a'|s')>0} \widehat{Q}^*(s', a')$ can be bounded by $\|\Delta_t\|_\infty$ with some constant. Thus, we have

$$Var[F_t(s)|\mathcal{F}_t] \leq C(1 + \|\Delta_t\|_\infty^2),$$

for some constant $C > 0$ under supremum norm. Thus, by Theorem B.2, $\Delta_t$ converges to zero w.p.1, i.e., $\widehat{Q}_t$ converges to $\widehat{Q}^*$ w.p.1. □

**Lemma B.4.** *The value estimation obtained by Equation* (21) *lower-bounds the value estimation obtained by Equation* (19)*:*

$$\widehat{Q}^*(s,a) - Q^*(s,a) \leq 0 \tag{27}$$

*for all state-action pairs.*

*Proof.* Following the definition of Equations (19) and (21), we can rewrite as

$$\max_{s,a}(\widehat{Q}^*(s,a) - Q^*(s,a))$$

$$= \max_{s,a}(\hat{\mathcal{B}}\widehat{Q}^*(s,a) - \mathcal{B}Q^*(s,a))$$

$$= \max_{s,a}(\sum_{s'\in\mathcal{S}} P(s'|s,a)[r + \gamma \max_{\hat{a}':\beta(\hat{a}'|s')>0} \widehat{Q}^*(s',\hat{a}')] - \sum_{s'\in\mathcal{S}} P(s'|s,a)[r + \gamma \max_{a'} Q^*(s',a')])$$

$$= \max_{s,a} \sum_{s'\in\mathcal{S}} P(s'|s,a)\gamma(\max_{\hat{a}':\beta(\hat{a}'|s')>0} \widehat{Q}^*(s',\hat{a}') - \max_{a'} Q^*(s',a'))$$

$$\leq \max_{s,a} \sum_{s'\in\mathcal{S}} P(s'|s,a)\gamma(\max_{\hat{a}'} \widehat{Q}^*(s',\hat{a}') - \max_{a'} Q^*(s',a'))$$

$$\leq \max_{s,a} \sum_{s'\in\mathcal{S}} P(s'|s,a)\gamma \max_{\tilde{a}}(\widehat{Q}^*(s',\tilde{a}) - Q^*(s',\tilde{a}))$$

$$\leq \max_{s,a} \gamma \sum_{s'\in\mathcal{S}} P(s'|s,a) \max_{\tilde{s},\tilde{a}}(\widehat{Q}^*(\tilde{s},\tilde{a}) - Q^*(\tilde{s},\tilde{a}))$$

$$= \gamma \max_{\tilde{s},\tilde{a}}(\widehat{Q}^*(\tilde{s},\tilde{a}) - Q^*(\tilde{s},\tilde{a})) = \gamma \max_{s,a}(\widehat{Q}^*(s,a) - Q^*(s,a))$$

where the last line follows from $\sum_{s'\in\mathcal{S}} P(s'|s,a) = 1$. Then we have

$$\max_{s,a}(\widehat{Q}^*(s,a) - Q^*(s,a)) \leq \gamma \max_{s,a}(\widehat{Q}^*(s,a) - Q^*(s,a))$$

$$\leq \gamma^2 \max_{s,a}(\widehat{Q}^*(s,a) - Q^*(s,a))$$

$$\leq \cdots$$

$$\leq \gamma^n \max_{s,a}(\widehat{Q}^*(s,a) - Q^*(s,a))$$

Take limit for both sides and since $0 < \gamma < 1$, we have $\max_{s,a}(\widehat{Q}^*(s,a) - Q^*(s,a)) \leq 0$.

When $\beta(a|s) > 0$ for all state-action pairs, the two contraction operators $\hat{\mathcal{B}}$ and $\mathcal{B}$ are the same. And based on Banach's fixed-point theorem, there is a unique fixed point. Thus $\widehat{Q}^*(s,a) = Q^*(s,a)$ for all state-action pairs., i.e., $\widehat{Q}^*(s,a) - Q^*(s,a) = 0, (s,a) \in \mathcal{S} \times \mathcal{A}$ holds when $\beta(a|s) > 0$ for all state-action pairs. □

Then, we get Theorem 4.1 proved with Lemmas B.1, B.3 and B.4.

## C  Experimental Details

In this section, we outline the detailed implementation settings, including basic PbRL configurations, feedback setup in experiments, and other training details.

### C.1  Preference-based RL Basic Settings

In this section, we provide additional details on unsupervised exploration and the uncertainty-based sampling scheme, as mentioned in Section 5.1. These techniques are crucial for improving feedback efficiency in algorithms, as referenced in Lee et al. [24]. To ensure a fair comparison, all preference-based RL algorithms in our experiments incorporate both unsupervised exploration and uncertainty-based sampling.

**Unsupervised Exploration.** Unsupervised exploration in preference-based RL, introduced by Lee et al. [24], involves designing an intrinsic reward based on the entropy of the state, effectively

encouraging the agent to explore diverse states and generate varied behaviors. Specifically, it utilizes a variant of particle-based entropy [32] as a computationally convenient entropy estimation method.

**Uncertainty-based Sampling.** Several sampling schemes exist, including uniform sampling, disagreement sampling, and entropy sampling. The latter two are categorized as uncertainty-based sampling and have demonstrated superior performance compared to uniform sampling, both intuitively and empirically. In our experiments, all methods (in the online settings) employ disagreement sampling schemes.

## C.2 Feedback in Experiments

**Number of Feedback.** In the **online setting**, we use 100 preference pairs for Cheetah Run, Button Press, and Window Open; 200 pairs for Walker Walk; 300 pairs for Push-5×5-1, Push-6×6-1, and Push-7×7-1; 1000 pairs for Quadruped Walk, Push-5×5-2, Push-6×6-2, Push-7×7-2, Strip-shaped Building, Block-shaped Building, and Simple Two-Story Building; and 4000 pairs for Sweep Into.

In the **offline setting**, we use 100 preference pairs for antmaze-medium-play-v2, antmaze-medium-diverse-v2, hopper-medium-expert-v2, walker2d-medium-expert-v2, can-ph, and lift-ph; 500 pairs for hopper-medium-replay-v2, walker2d-medium-replay-v2, can-mh, and lift-mh; and 1000 pairs for antmaze-large-play-v2 and antmaze-large-diverse-v2.

**Human Labels.** We incorporate human feedback from subjects with prior experience in robotic tasks, as outlined in PT [21]*. An informed human annotator evaluates each task by watching video clips of trajectory segments and selecting the one that more effectively achieves the task objective. Each segment lasts 3 seconds (equivalent to 100 time steps). If no clear preference is observed, the annotator may assign equal preference to both segments by selecting a neutral option.

## C.3 Architecture and hyperparameters.

In this section, we describe the architecture of the neural networks used in the SAC algorithm, which serves as the baseline method. The actor in SAC consists of two layers with 1024 hidden units. The two Q networks in SAC share the same architecture as the actor. This consistent architecture helps streamline the optimization process across the actor and critic networks. The detailed neural network parameters and hyperparameters for SAC are shown in Table 8a. Additionally, Table 8b presents the distinct hyperparameters for PEBBLE and STAR.

| Hyperparameter | Value | Hyperparameter | Value |
|---|---|---|---|
| Discount | 0.99 | Batch size | 1024 |
| $(\beta_1, \beta_2)$ | (0.9,0.999) | Initial temperature | 0.1 |
| Hidden layer units | 1024 | Critic target update freq | 2 |
| Activation Function | ReLU | Critic $\tau$ | 0.005 |
| Critic optimizer | AdamW | Learning rate | 5e-4 |
| Actor optimizer | AdamW | | |

(a) Hyperparameters of SAC.

| Hyperparameter | Value | Hyperparameter | Value |
|---|---|---|---|
| Length of segment | 50 | Reward model size | 3 |
| Learning rate | 0.0003 | Frequency of feedback | 5000 |
| Reward batch size | 128 | Number of train steps | 1e6 |
| Reward update | 200 | Replay buffer capacity | 1e6 |
| Scaling parameter $\beta$ (STAR) | 6 | Graph update batch size (STAR) | 32 |
| margin parameter $\varepsilon$ (STAR) | 0.05 | Regularizer weight $\eta$ (STAR) | 0.1 |

(b) Hyperparameters of PEBBLE and STAR.

# D   Environment Specifications

In this section, we delineate the tasks utilized in our experiments. For online settings, discrete tasks include Sokoban [42] and Craftenv [71], while continuous tasks encompass robotic manip-

---

*https://github.com/csmile-1006/PreferenceTransformer

ulation challenges from Meta-world [63] and locomotion tasks from the DeepMind Control Suite (DMControl) [50, 52]. Regarding offline settings, we incorporate control tasks from the D4RL benchmarks [7].

## D.1 Sokoban

Sokoban [42], the Japanese word for 'a warehouse keeper', is a puzzle video game, which is analogous to the problem of having an agent in a warehouse push some specified boxes from their initial locations to target locations. Target locations have the same number of boxes. The goal of the game is to manipulate the agent to move all boxes to the target locations. Specifically, the game is played on a rectangular grid called a room, and each cell of the room is either a floor or a wall. At each new episode, the environment will be reset, which means the layout of the room is randomly generated, including the floors, the walls, the target locations, the boxes' initial locations, and the location of the agent. We choose six tasks with different complexities from Push-5×5-1 to Push-7×7-2, which is shown in Figure 8. The numbers in the task name denote respectively the size of the grid and the number of boxes.

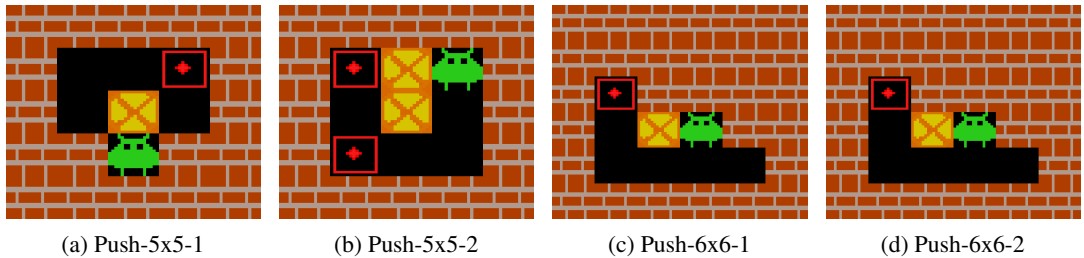

| (a) Push-5x5-1 | (b) Push-5x5-2 | (c) Push-6x6-1 | (d) Push-6x6-2 |

Figure 8: Visualization of puzzle tasks from Sokoban, which focuses on evaluating the capabilities of agents in spatial reasoning, logical deduction, and long-term planning.

**State Space.** The state space consists of all possible images displayed on the screen. Each image has the same size as the map, and using the way of dividing each pixel of the image by 255 to normalize into [0,1], we preprocess the image before inputting.

**Action Space.** The action space of Sokoban has a total of eight actions, composed of moving and pushing the box in four directions, which are *left*, *right*, *up*, *down*, *push-left*, *push-right*, *push-up*, *push-down* in detail.

**Reward Setting.** The agent gets a punishment with a -0.1 reward after each time step. Successfully pushing a box to the target location, can get a +1 reward, and if all boxes are laid in the right locations, the agent can obtain an extra +10 reward. We set the max episode steps to 120, which means the cumulative reward during one episode ranges from -12 to 10 plus the number of boxes.

## D.2 CraftEnv

Craftenv [71], A Flexible Robotic Construction Environment, is a collection of construction tasks. The agent needs to learn to manipulate the elements, including smartcar, blocks, and slopes, to achieve a target structure through efficient and effective policy. Each construction task is a simulation of the corresponding complex real-world task, which is challenging enough for reinforcement learning algorithms. Meanwhile, the CraftEnv is highly malleable, enabling researchers to design their own tasks for specific requirements. The environment is simple to use since it is implemented by Python and can be rendered using PyBullet. We choose three different designs of the building tasks, shown in Figure 9, to evaluate our algorithm in CraftEnv.

**State Space.** We assume that the agent can obtain all the information in the map. Therefore, the state consists of all knowledge of smartcar, blocks, folded slopes, unfolded slopes' body, and unfolded slopes' foot, including the position and the yaw angle.

**Action Space.** The available actions of an agent are designed based on real-world smartcar models, including a total of fifteen actions. Besides all eight directions moving actions, i.e. *forward*, *backward*, *left*, *right*, *left-forward*, *left-backward*, *right-forward*, and *right-backward*, there are interaction-related actions, designed to simulate the building process in the real world. Specifically, the agent can act *lift*

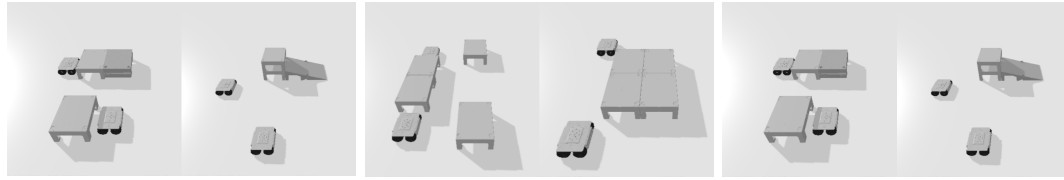

(a) The Strip-shaped Building     (b) The Block-shaped Building     (c) The Simple Two-Story Building

Figure 9: Visualization of building tasks from CraftEnv. From left to right are The Strip-shaped Building, The Block-shaped Building, and The Simple Two-Story Building task respectively.

and *drop* actions to decide whether or not to carry the surrounding basic element, and can *flod* or *unflod* slopes to build the complex buildings. In addition, the actions of *rotate-left* and *rotate-right* control the agent to rotate the main body to the left and right, and *stop* action is just a non-action.

**Reward Setting.** CraftEnv is a flexible environment as mentioned above. We can specify our own reward function for different construction tasks. For the relatively simple tasks of building with specified shape requirements, we can use discrete reward, where some reward is given when part of the blueprint is built. While, for building tasks with high complexity, various reward patterns should be designed to encourage the agent to build with different intentions.

### D.3 Robotic tasks

In our experiments, we utilize robotic manipulation tasks from Meta-world [63] and locomotion tasks from the DeepMind Control Suite (DMControl) [50, 52], which is visualized in Figure 10. Meta-World comprises 50 diverse manipulation tasks with a common structural framework, while DMControl offers a collection of stable, rigorously tested tasks, serving as benchmarks for continuous control learning agents. These tasks are equipped with well-formulated reward functions that facilitate agent learning. For performance evaluation, Meta-world introduces an interpretable success metric for each task, which serves as the standard evaluation criterion across various settings. This metric often hinges on the proximity of task-relevant objects to their final goal positions, quantified as $||o - g|| \leq \epsilon$, with $\epsilon$ representing a nominal distance threshold, such as 5 cm. DMControl, conversely, employs episode returns for evaluation. We adopt fixed-length episodes of 1000 timesteps as a proxy. Given that all reward functions are designed such that $r \approx 1$ is in proximity to goal states, learning curves that measure total returns can maintain consistent y-axis limits of $[0, 1000]$. This uniformity simplifies interpretation and facilitates averaging across tasks.

### D.4 D4RL benchmark

**AntMaze**. AntMaze involves a navigation challenge where a Mujoco Ant robot must locate and reach a specified goal. The data for this task is derived from a pre-trained policy designed to navigate various maze layouts, primarily focusing on two configurations: *medium* and *large*. The data are compiled using two distinct strategies: *diverse* and *play*. *Diverse* data sets are produced by the pre-trained policy, which utilizes random start and goal locations, whereas *play* data sets are generated with specific, intentionally selected goal locations. The task's reward structure is sparse, awarding points only when the robot is within a predefined proximity to the goal; otherwise, no reward is granted.

**Gym-Mujoco Locomotion**. In Gym-Mujoco locomotion tasks, the objective is to manage simulated robots (Walker2d, Hopper) to advance forward while minimizing energy expenditure (action norm) to ensure safe behavior. Two data generation methodologies are employed: *medium-expert* and *medium-replay*. The emphmedium-expert data sets have an equal mix of expert demonstrations and suboptimal (partially-trained) demonstrations. The emphmedium-replay data sets come from the replay buffer of a partially-trained policy. The robot's forward velocity, a control penalty, and a survival bonus all contribute to determining the task's reward.

**Robosuite Robotic Manipulation**. In Robosuite's robotic manipulation tasks [74], various 7-DoF simulated hand robots perform distinct tasks. Our experiments utilize environments simulated with Panda by Franka Emika, focusing on two specific tasks: lifting a cube (*lift*) and relocating a coke can from a table to a designated bin (*can*). Data collection involves inputs from either a single proficient teleoperator (*ph*) or six teleoperators with varying skill levels (*mh*). The task's reward is sparse, with further details deferred to the original paper.

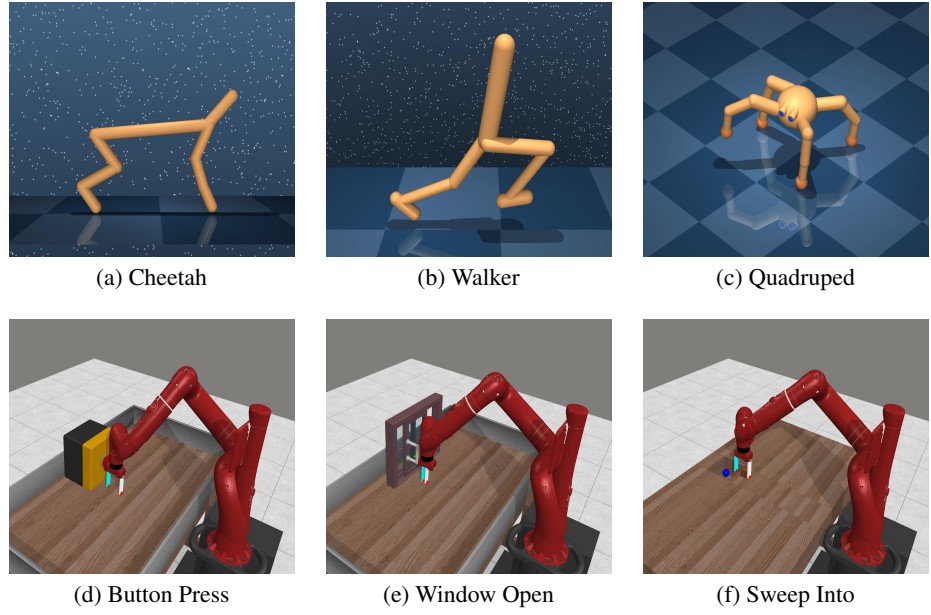

|  |  |  |
|:-:|:-:|:-:|
| (a) Cheetah | (b) Walker | (c) Quadruped |
| (d) Button Press | (e) Window Open | (f) Sweep Into |

Figure 10: Six tasks are used for experiments. (a-c) DMControl tasks. (d-f) Meta-world tasks.

## E   Full Experiments

### E.1   Results on Discrete Settings

A detailed introduction and visualization of the six puzzle-solving tasks from Sokoban and the three flexible environments from CraftEnv are provided in Appendix D. We selected these tasks to cover a range of complexities for our experiments. Figure 11 shows the learning curves of the average episode return for STAR and the baselines across discrete tasks. In each task, SAC, utilizing the ground-truth reward, serves as the upper performance bound. As seen in Figure 11, STAR demonstrates rapid performance gains early in training across various tasks. Remarkably, in several tasks, STAR achieves close to SAC's performance while using significantly fewer human preference labels, indicating high feedback efficiency. For example, in the Strip-shaped building task, STAR surpasses PEBBLE's average performance using only 30% of the total samples, highlighting its sample efficiency. In contrast, some baselines, affected by randomness in challenging tasks, show a decline in performance as training progresses. This underscores STAR 's robustness in maintaining performance even with fewer feedback labels. These results demonstrate that STAR substantially reduces the amount of feedback required to tackle complex tasks effectively, making it a highly efficient approach in PbRL.

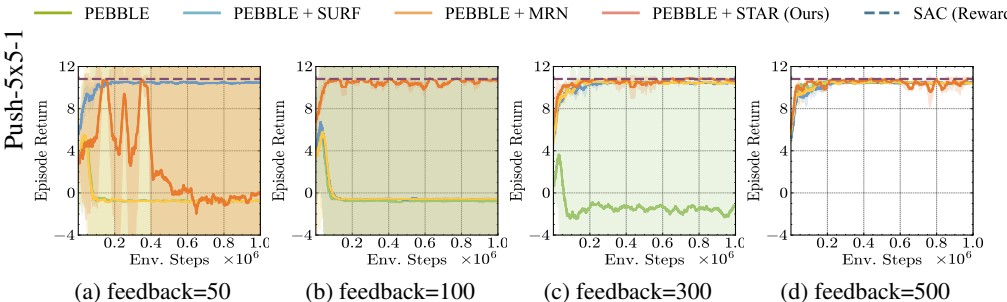

| (a) feedback=50 | (b) feedback=100 | (c) feedback=300 | (d) feedback=500 |

Figure 12: Training curves of all methods with varying numbers of preference labels on Push-5x5-1. The solid line presents the mean values, and the shaded area denotes the standard deviations over five runs.

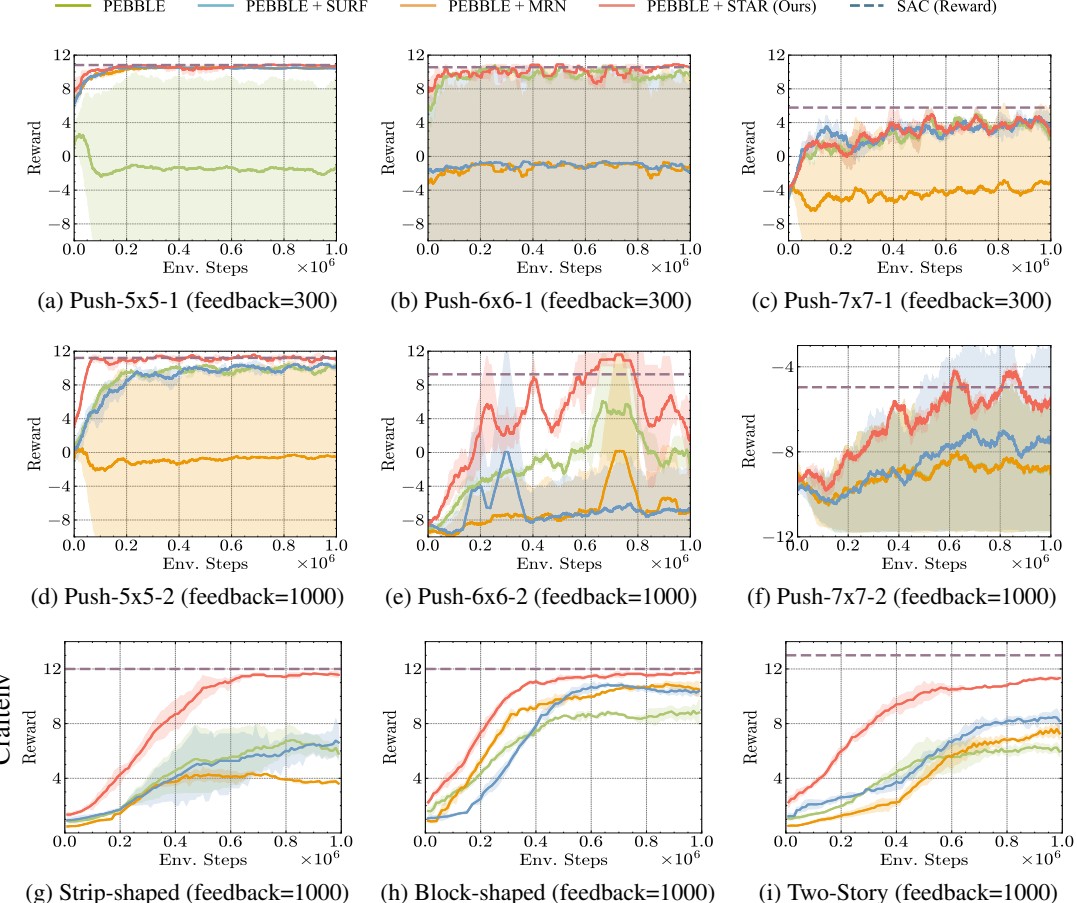

Figure 11: Evaluating curves of all methods on locomotion tasks and robotic manipulation tasks. The solid line presents the mean values, while the shaded area indicates the 78% confidence interval over five runs. The red line is our method.

### E.2 Ablation Studies

**Impact of Preference Quantity.** To assess how the amount of human preferences affects STAR 's performance, we conducted an additional experiment using varying numbers of preference labels: $N \in 50, 100, 300, 500$ on Sokoban tasks. The training curves of the average episode return for all methods are shown in Figure 12. The results reveal a clear trend: as the number of preference labels increases, STAR 's policy performance steadily improves. Notably, with sufficient preference labels, STAR approaches the performance upper bound set by SAC, highlighting its efficiency in leveraging feedback data. This trend suggests that while STAR performs well with limited preferences, providing more feedback enables it to closely match optimal performance. However, as the learning curves indicate, the rate of improvement begins to taper off at higher preference counts, suggesting diminishing returns beyond a certain threshold.

**Computational Efficiency of STAR.** When deploying algorithms in real-world applications, training time overhead is a critical factor. In discrete settings, we store vertices in a dictionary using hashes as keys, enabling efficient state retrieval in $\mathcal{O}(1)$ time. Additionally, updates to the graph are only required at the end of each episode, further minimizing computational overhead. In continuous settings, STAR reduces computational costs by allowing the actor to update at a lower frequency compared to traditional methods. To quantitatively assess the time overhead, Table 5 summarizes the training times (in hours) for STAR and baseline methods on Push-7x7-2 and Cheetah Run. For instance, STAR demonstrates only a slight increase in training time compared to PEBBLE, with an overhead of less than 3%. In contrast, SURF incurs a noticeable average increase of approximately 23.4% due to the additional time required for labeling samples. These results demonstrate that STAR

Table 4: Training time (hour) of various tasks. Each run is conducted using the exact same hardware environment.

| Task/Methods | PEBBLE | SURF | MRN | STAR (ours) |
|---|---|---|---|---|
| Push-7x7-2 | 2.78 | 3.56 (+28.1%) | 3.11 (+11.9%) | 2.84 (**+2.2%**) |
| Cheetah Run | 3.23 | 3.83 (+18.6%) | 3.42 (+5.9%) | 3.31 (**+2.5%**) |

Table 5: Parameter search for policy regularization.

| $\eta$ | 0 | 0.1 | 0.2 | 0.5 | 1 |
|---|---|---|---|---|---|
| Quadruped Walk | 790 | 763 | 728 | 643 | 620 |
| Button Press | 47 | 50 | 39 | 28 | 24 |
| Sweep Into | 55 | 58 | 52 | 49 | 48 |
| Cheetah Run | 710 | 713 | 697 | 579 | 542 |

maintains a favorable balance between performance and training efficiency without significantly increasing computational costs.

**Sensitivity Analysis of the Parameter** $\eta$**.** Policy regularization is a crucial component of STAR. In our experiments, we performed a parameter search for $\eta$ and selected a stable value within the mid-range of the search space. The table below presents the performance of STAR on Cheetah Run for different values of $\eta$. For all experiments conducted in online settings, we set $\eta$ to 6, as shown in Table 8b.

