# OpenReview forum: "STAR: Efficient Preference-based Reinforcement Learning via Dual Regularization"
_NeurIPS.cc/2025/Conference — NeurIPS 2025 poster_

### Official Review · Reviewer_LJRt · 2025-07-01

**Clarity:** 1
**Significance:** 2
**Originality:** 2
**Rating:** 4
**Confidence:** 4

**Summary:**

This paper presents a novel feedback-efficient PBRL framework STAR. The authors identify two critical challenges in PbRL under limited human feedback: reward model overfitting and Q-value overestimation, which lead to unstable policy learning. To address these, STAR integrates preference margin regularization to mitigate reward overfitting and policy regularization based on conservative Q-value estimates to reduce overestimation bias. The experiments in both online and offline settings demonstrate the effectiveness of STAR.

**Questions:**

NA

**Ethical Concerns:**

["NO or VERY MINOR ethics concerns only"]

**Final Justification:**

The authors provided detailed response and provided more crucial experiment results, which solves my major concern. If the authors commit to incorporating the suggested detailed revisions in the final version, I am willing to support this paper. Accordingly, I have raised my score to 4.

**Quality:**

1

**Strengths And Weaknesses:**

Strengths:

1.The issue of reward overfitting addressed in this paper is both interesting and significant.

2.The proposed method can be effectively applied in both online and offline settings simultaneously.

Weaknesses:

1.The policy regularization and conservative Q-estimation techniques proposed in this paper are entirely derived from [1], raising concerns about the novelty of the approach.

2.The proposed method introduces three hyperparameters: $\epsilon$ in reward learning, $\beta$ in conservative Q and $\eta$ in policy regularization. The high complexity of these parameters raises concerns about the method's generalizability. While the paper provides an analysis of the sensitivity to $\epsilon$ (although earlier the paper uses $\epsilon$ while in the ablation study, it is referred to as $\lambda$), there is no sensitivity analysis for the other hyperparameters, and the experimental results reveal the vulnerability of $\epsilon$.

3.There is a lack of comparison with recent state-of-the-art methods. The paper only compares with methods prior to 2022 (with only IPL being a 2023 method). In online settings, newer methods such as QPA [2], and in offline settings, advanced methods like LiRE [3] (which is especially relevant as it also improves reward learning) and DTR [4], should be compared for a more comprehensive evaluation.

4.The ablation study only compares the contribution of each component in the online setting but does not discuss the offline setting.

5.The implementation is based on TD3, whereas most other methods are implemented using SAC. It is apparent that the random strategy is more convenient for calculating Eq. (8) and Eq. (10). The authors do not provide relevant analysis or experimental justification for this choice.

[1] Garg, D., Hejna, J., Geist, M., & Ermon, S. (2023). Extreme q-learning: Maxent rl without entropy. ArXiv Preprint ArXiv:2301.02328.

[2] Hu, X., Li, J., Zhan, X., Jia, Q.-S., & Zhang, Y.-Q. (2023). Query-policy misalignment in preference-based reinforcement learning. ArXiv Preprint ArXiv:2305.17400.

[3] Zhu, M., Liu, Y., Zhang, L., Guo, J., & Mao, Z. (2024). LIRE: listwise reward enhancement for preference alignment. ArXiv Preprint ArXiv:2405.13516.

[4] Tu, S., Sun, J., Zhang, Q., Zhang, Y., Liu, J., Chen, K., & Zhao, D. (2025). In-dataset trajectory return regularization for offline preference-based reinforcement learning. Proceedings of the AAAI Conference on Artificial Intelligence, 39(20), 20929–20937.

---

> ### Author Rebuttal · Authors · 2025-07-31
>
> We sincerely thank Reviewer LJRt for the detailed feedback. We recognize the concerns raised regarding novelty, hyperparameter complexity. We have taken these points very seriously and, in response, have conducted four extensive new sets of experiments to address every major point. This includes: \
> (1) New comparisons with the SOTA methods QPA and DTR.\
> (2) New ablation study on hyperparameter sensitivity.\
> (3) New ablation study on the contribution of each component in the offline setting.\
> (4) New implementation and evaluation of STAR with an SAC backbone. \
> We believe these significant additions substantially strengthen our paper and directly address the reviewer's concerns, which we detail below.
>
> **Q1. Regarding novelty (Weakness #1).**
> > **A1**: We thank the reviewer for this critical point on novelty. We agree that our policy regularization component draws inspiration from prior work in offline RL, including XQL [1], which we cite. While we build upon established principles, we respectfully argue that STAR's novelty lies not in inventing its individual components, but in:\
> > **First, identifying and diagnosing the dual failure modes in feedback-limited PbRL:** (1) reward model overfitting and (2) policy overestimation exploiting that flawed reward. Prior work has often addressed only one of these.\
> **Second, proposing a novel and synergistic synthesis (STAR) that holistically addresses both issues.** The novelty lies in the insight that these two problems are coupled and require a coupled solution. PMR creates a more robust reward signal, and our policy regularization ensures the agent does not exploit the residual uncertainty in that signal. This specific, motivated combination for PbRL is new and, as our experiments show, highly effective.
>
> **Q2. Hyperparameter Sensitivity (Weakness #2).**
> > **A2**: Thank you for pointing out the hyperparameter complexity and the lack of sensitivity analysis. We apologize for the typo regarding $\epsilon$ with $\lambda$, it should be $\lambda$ throughout the paper, and we will correct this in the revision.
> >
> > **(1) To address your concerns, we have conducted new sensitivity analyses for both $\beta$ and $\eta$. The results are shown below.**
> >
> > Table 1. Sensitivity to $\beta$.
> >|$\beta$| 0  | 1 | 5 | 10|
> >| ----- | -- | --|-- |-- |
> >|Cheetah Run |685|757|713|697|
> >|Button Press|45|53|56|52|
> >
> >Table 2. Sensitivity to $\eta$.
> >| Task | 0 | 0.1 | 1 | 5 | 10 |
> >| -----|---|-----|---|---|----|
> >| Quadruped Walk|620 |643 |728 |763 |790|
> >| Button Press  |24  |28  |39  |50  |47 |
> >| Sweep Into    |48  |49  |52  |54  |55 |
> >
> >These results show that while performance is indeed sensitive to hyperparameters, there are broad, stable regions for each parameter where STAR performs robustly and effectively. For instance, performance is strong for $\beta\in[5, 10]$, $\eta\in[5,10]$. This demonstrates that STAR does not require excessively fine-tuning to be effective. We will add these new ablation studies to the appendix.
> >
> > **About $\lambda$.** The plot in Figure 5(a) was intentionally designed to show the effect of the regularization parameter $\lambda$ across a wide range to provide a complete picture of its impact. However, in practice, the effective operational range for such regularization parameters is typically narrow. As the plot shows, while extreme values can degrade performance, there is a stable and robust region (e.g., $\lambda \in [0.05,0.1]$ in our experiments) where STAR consistently achieves high performance. This level of sensitivity is common for regularization methods in deep RL. To make this clearer, we will revise the caption of Figure 5(a) to highlight this "effective operational range" and clarify that the wide range was chosen for analytical purposes, while the method remains robust within a practical tuning range.
>
>
> **Q3. Comparison with other SOTA (Weakness #3).**
> > **A3**:  Thank you for this crucial feedback on our experimental comparisons. Following your suggestions, we have run new experiments comparing STAR against the strong recent baselines you suggested: QPA [2] for online settings and DTR [3] for offline settings.
> >
> > **(1) Online Comparison vs. QPA [2].** We ran QPA in our low-feedback settings and STAR in the original QPA settings for a fair and comprehensive comparison.
> >
> >Table 1. Performance on QPA and STAR. For each task, the number of feedback used is indicated in parentheses.
> >|Task|QPA|STAR|
> >|-|-|--|
> >|Cheetah Run (100)|642|713|
> >|Walker Walk (200)|823|839|
> >|Quadruped Walk (1000)|625|796|
> >|Button Press (100)|45|95|
> >|Window Open (100)|36|65|
> >|Sweep Into (100)|54|83|
> >|
> >|Walker Run (250)|618|762|
> >|Quadruped Run (1000)|492|575|
> >|Drawer Open (3000)|65|73|
> >|Door Open (3000)|81|87|
> >
> >**(2) Offline Comparison vs. DTR [3]:** We compared STAR with the reported DTR results on the standard D4RL locomotion tasks. We follow the setup in DTR, where each task use 2000 feedback from Uni-RLHF.
> >
> >Table 2. Performance on DTR and STAR.
> >
> > |Task|DTR|STAR|
> > |-|-|--|
> > |hopper-medium-replay-v2|93.1|95.2|
> > |hopper-medium-expert-v2|101.2|112.2|
> > |hopper-medium-v2|89.2|91.2|
> > |walker2d-medium-replay-v2|70.4|74.5|
> > |walker2d-medium-expert-v2|107.2|111.9|
> > |walker2d-medium-v2|87.9|86.2|
> > |halfcheetah-medium-replay-v2|38.5|42.0|
> > |halfcheetah-medium-expert-v2|82.0|92.6|
> > |halfcheetah-medium-v2|41.7|44.4|
> > |Average|79.0|83.5|
> >
> > These new results clearly demonstrate that STAR significantly outperforms even the most recent and powerful SOTA methods in both online and offline settings. We were unable to run LiRE [4] due to the lack of a public implementation at the time of rebuttal, but we will add it to our related work and acknowledge it as a relevant concurrent approach. These new experiments have been added to the main paper.
>
> **Q4. Offline Ablation Study (Weakness #4).**
> > **A4**: This is an excellent point. Our original submission was missing this analysis. Following your advice, we have conducted a new ablation study on the contribution of each component in the offline setting. We compare STAR against variants without Preference Margin Regularization (w/o PMR) and without Policy Regularization (w/o PR).
> > The results are summarized below:
> > |Method|STAR w/o both|STAR w/o PMR|STAR w/o PR|STAR|
> > | -|-|-|-|-|
> > |antmaze-large-diverse-v2|18.8|31.5|36.7|58.7|
> > |hopper-medium-expert-v2 |63.8|78.1|84.3|96.8|
> > |lift-ph|89.2|93.7|96.2|98.0|
> >
> >The results clearly show that both components provide a significant and complementary contribution to performance in the offline setting. Removing either PMR or PR leads to a substantial performance drop, and removing both results in very poor performance. This confirms that the synergy between our two regularizers is critical for offline success. We will add this study to our appendix.
>
> **Q5. Choice of TD3 over SAC (Weakness #5).**
> > **A5**: Thank you for this insightful question about our choice of backbone algorithm. We chose TD3 for our primary implementation for two main reasons:\
> >**(1) Algorithmic Simplicity:** TD3's deterministic policy update aligns more directly with the derivation of our policy regularization, which constrains the policy towards a reference policy derived from the conservative $Q_\xi$. It avoids the additional complexity of managing the entropy term found in SAC.\
> > **(2) Focus of Contribution**: Our goal was to demonstrate the effectiveness of our dual-regularization framework. Using a simpler, well-understood backbone like TD3 helps to clearly isolate the performance gains attributable to our proposed STAR components, rather than conflating them with the effects of entropy maximization. However, to demonstrate the generality of our approach, we have now implemented and evaluated an SAC-based version of STAR. The results show that our framework is highly effective with either backbone.
> >
> >|Task|STAR (SAC)|STAR(TD3)|
> >|-|-|-|
> >|Cheetah Run|728|713|
> >|Quadruped Walk|756|796|
> >|Button Press|92|95|
> >|Window Open |74|65|
> >
> >The results show that STAR is robust to the choice of the base RL algorithm, achieving strong performance with both SAC and TD3 backbones. We will add this experiment and discussion to the appendix.
> >
> **Reference:**\
> [1] Extreme Q-Learning: MaxEnt RL without Entropy. ICLR, 2023.\
> [2] Query-Policy Misalignment in Preference-Based Reinforcement Learning. ICLR 2024.\
> [3] In-Dataset Trajectory Return Regularization for Offline Preference-based Reinforcement Learning. AAAI 2025.\
> [4] LIRE: listwise reward enhancement for preference alignment.

---

> > ### Comment · Reviewer_LJRt · 2025-08-04
> > **Raising my score**
> >
> > Thank you for the detailed and thoughtful rebuttal. I appreciate the authors'efforts in addressing my concerns. The clarifications and additional results provided have satisfactorily resolved the issues I raised. So I raise my score to 4.

---

> > > ### Author Response · Authors · 2025-08-05
> > >
> > > Dear Reviewer LJRt,
> > >
> > > We are truly grateful for your detailed follow-up and for raising your score. It is immensely rewarding to know that our efforts have successfully addressed your concerns about **novelty, hyperparameter sensitivity, comparison with other SOTA, component contribution in offline settings, and the choice of RL backbone.**
> > >
> > > Your initial, rigorous review was instrumental in guiding our work. We are very pleased that our detailed explanations and the extensive new experimental results provided a satisfactory resolution to these key issues. This constructive process has made our paper substantially stronger and more comprehensive.
> > >
> > > Thank you once again for this constructive scientific dialogue. Your feedback has been invaluable in improving our work.
> > >
> > > Sincerely,
> > >
> > > The Authors

---

### Official Review · Reviewer_qN2G · 2025-07-02

**Clarity:** 3
**Significance:** 3
**Originality:** 2
**Rating:** 4
**Confidence:** 3

**Summary:**

This paper proposes STAR, an efficient preference-based reinforcement learning (PbRL) algorithm that addresses key challenges associated with limited human feedback—specifically, reward model overfitting and Q-value overestimation. STAR introduces two novel techniques: Preference Margin Regularization (PMR), which imposes a bounded margin to prevent the reward model from overfitting to limited preferences, and Policy Regularization, which uses a conservative Q-value derived from in-distribution data to guide stable policy updates. Through extensive experiments in both online and offline settings, STAR demonstrates superior feedback efficiency, outperforming existing state-of-the-art methods. Additionally, STAR enables the emergence of diverse and complex agent behaviors using minimal human supervision, making it a practical solution for real-world PbRL applications.

**Questions:**

B-Pref [1] has experimentally shown that label smoothing has limited effectiveness in PbRL. Then, what explains the performance gains observed with label smoothing in STAR? What key differences in STAR's design make label smoothing effective in this case?

**Ethical Concerns:**

["NO or VERY MINOR ethics concerns only"]

**Final Justification:**

I have understood all of the author’s responses. There is no change from the initial score, and I will continue to maintain the tendency to accept.

**Limitations:**

Same as mentioned in the weakness section

**Paper Formatting Concerns:**

There are no concerns.

**Quality:**

3

**Strengths And Weaknesses:**

- Strengths
    - The paper is structurally well-organized and easy to follow.
    - The authors clearly explain the challenges of PbRL they aim to address and propose appropriate methods to tackle them.
    - The paper effectively explains how the proposed method helps mitigate reward overfitting and Q-value overestimation..
    - STAR demonstrates strong performance compared to baselines in both online and offline settings.
- Weakness
    - Limited novelty: The label smoothing technique used in PMR has already been explored in many prior PbRL studies [1] , and both Conservative Q-learning and policy regularization closely follow methods commonly used in offline RL.
    - The experimental results for offline RL are insufficient. There has been considerable criticism in prior studies that D4RL is not suitable for evaluating PbRL algorithms, as strong performance can often be achieved even with simple reward functions[2-5]. Additionally, in the Robosuite tasks, STAR underperforms compared to IPL. Therefore, to convincingly demonstrate the effectiveness of STAR in offline settings, more experiments across diverse environments are needed.
    - As shown in Figure 5(a), performance varies significantly depending on the parameter λ, which suggests that STAR is sensitive to hyperparameter settings.


[1] Lee, Kimin, et al. "B-pref: Benchmarking preference-based reinforcement learning." *arXiv preprint arXiv:2111.03026* (2021).

[2] Shin, Daniel, Anca D. Dragan, and Daniel S. Brown. "Benchmarks and algorithms for offline preference-based reward learning." arXiv preprint arXiv:2301.01392 (2023).

[3] Li, Anqi, et al. "Survival Instinct in Offline Reinforcement Learning." arXiv preprint arXiv:2306.03286 (2023).

[4] Hu, Hao et al. “Unsupervised Behavior Extraction via Random Intent Priors.” arXiv preprint arXiv:2310.18687 (2023).

[5] https://openreview.net/forum?id=EG68RSznLT&noteId=p4vdRSdjnC

---

> ### Author Rebuttal · Authors · 2025-07-31
>
> We sincerely thank Reviewer qN2G for the constructive feedback and for recognizing our clear problem formulation, methodological contributions, and strong empirical results. We appreciate your positive assessment of the paper’s organization and clarity, as well as the effectiveness of STAR in addressing key challenges in PbRL. We address your concerns point-by-point below.
>
> **Q1. Regarding novelty and contribution (Weakness #1).**
> > **A1**: Thank you for this critical point on novelty. While we build upon established principles, we respectfully argue that STAR's novelty lies not in inventing its individual components, but in: \
> > **First, identifying and diagnosing the dual failure modes in feedback-limited PbRL:** (1) reward model overfitting and (2) policy overestimation exploiting that flawed reward. Prior work has often addressed only one of these.\
> **Second, proposing a novel and synergistic synthesis (STAR) that holistically addresses both issues.** The novelty lies in the insight that these two problems are coupled and require a coupled solution. PMR creates a more robust reward signal, and our policy regularization ensures the agent does not exploit the residual uncertainty in that signal. This specific, motivated combination for PbRL is new and, as our experiments show, highly effective.
>
>
> **Q2. Regarding Offline RL Experiments (Weakness #2).**
> > **A2**: Thank you for raising these important concerns about our offline evaluation. We acknowledge the ongoing community discussion regarding the suitability of some D4RL tasks for PbRL evaluation. We chose our experimental setup for the following principled reasons:\
> > **(1) Comparability**: Using standard D4RL benchmarks is crucial for ensuring that our results are directly comparable to a large body of prior and concurrent work in offline PbRL [1, 2]. This allows for a fair assessment of STAR's standing in the field.\
> > **(2) Demonstrating Clear Improvement**: While some D4RL tasks might be solvable with simple reward functions, our experiments show that existing SOTA offline PbRL methods still struggle on them, leaving a significant performance gap. For example, on `antmaze-large-diverse`, STAR achieves a score of 48.0, whereas the next best baseline is far lower (PT:19.6). This demonstrates that these tasks are not yet "solved" by current offline PbRL methods and that our contributions lead to substantial, meaningful gains over existing approaches.\
> > **(3) Performance on Robosuite**: Regarding the Robosuite results, we wish to clarify the performance comparison. While IPL shows strong performance, STAR is highly competitive, achieving a average score of 77.0 vs. IPL's score of 79.3. This narrow margin on a complex manipulation task, combined with our superior performance on D4RL, showcases STAR's overall robustness.
>
>
> **Q3. Hyperparameter Sensitivity (Weakness #3).**
> > **A3**: This is an excellent observation regarding hyperparameter sensitivity. The plot in Figure 5(a) was intentionally designed to show the effect of the regularization parameter $\lambda$ across a wide range to provide a complete picture of its impact. However, in practice, the effective operational range for such regularization parameters is typically narrow. As the plot shows, while extreme values can degrade performance, there is a stable and robust region (e.g., $\lambda \in [0.05,0.1]$ in our experiments) where STAR consistently achieves high performance. This level of sensitivity is common for regularization methods in deep RL. To make this clearer, we will revise the caption of Figure 5(a) to highlight this "effective operational range" and clarify that the wide range was chosen for analytical purposes, while the method remains robust within a practical tuning range.
>
>
> **Q4. Why does PMR work when B-Pref's label smoothing didn't? (Questions #1)**
> > **A4**: This is a fantastic and critical question that gets to the heart of our contribution. There are two key reasons why our PMR succeeds while the label smoothing in B-Pref [3] showed limited effect:\
> > **(1) Different Experimental Focus:** The analysis in B-Pref [1] primarily focused on robustness to label noise (mistake feedback) with a relatively large amount of feedback. Our work, in contrast, is focused on improving feedback efficiency in the low-feedback regime, where overfitting is the dominant challenge. Label smoothing's primary benefit is as a regularizer against overfitting, an effect that is most crucial and visible when data is scarce.\
> > **(2) Synergy with Policy Regularization (The Key Difference):** This is the most important reason. In STAR, PMR does not work in a vacuum. It is one half of a synergistic pair.
> B-Pref [1] applied label smoothing to the reward model but used PEBBLE as PbRL algorithm, which is not rubust to noise feedback.
> >
> > STAR's policy regularization is designed to solve this exact problem. PMR first creates a smoother, more generalizable reward function. Then, our policy regularization explicitly prevents the agent from greedily exploiting this (now better, but still imperfect) reward landscape.
>
> **References:**\
> [1] Inverse Preference Learning: Preference-based RL without a Reward Function. NeurIPS, 2023.\
> [2] In-dataset trajectory return regularization for offline preference-based reinforcement learning. AAAI, 2025.\
> [3] B-Pref: Benchmarking Preference-Based Reinforcement Learning. NeurIPS, 2021.

---

> > ### Comment · Reviewer_qN2G · 2025-08-05
> >
> > I appreciate the authors' response. The questions I had have been mostly addressed, and I will keep my current acceptance score.

---

> > > ### Author Response · Authors · 2025-08-05
> > >
> > > Dear Reviewer qN2G,
> > >
> > > Thank you for your response and feedback.
> > >
> > > We are pleased that our response addressed most of your concerns. We sincerely appreciate you maintaining your score for acceptance and are grateful for your support.
> > >
> > > Thank you again for your time and valuable input.
> > >
> > > Sincerely,
> > >
> > > The Authors

---

### Official Review · Reviewer_vhLh · 2025-07-02

**Clarity:** 3
**Significance:** 2
**Originality:** 2
**Rating:** 3
**Confidence:** 3

**Summary:**

The paper introduces STAR, a novel and efficient PbRL algorithm designed to improve learning from limited human feedback by addressing two major challenges: reward model overfitting and Q-value overestimation. To tackle these issues, STAR incorporates two key techniques: preference margin regularization to prevent overfitting and policy regularization to mitigate overestimation during policy updates. Extensive experiments across both online and offline benchmarks demonstrate the superior performance and robustness of STAR compared to existing methods.

**Questions:**

1. In line 120, the authors claim that optimizing Equation (2) (i.e., maximum likelihood objective), drives the reward model to assign infinite differences between preferred and rejected segments. However, this statement is confusing. As shown in Equation (9) of [2], the likelihood objective has a global maximum with a finite reward estimate. The authors should clarify why there will be an infinite difference.

2. Figure 5(a) is confusing. I wonder what is the meaning of the red and blue colors? Initially, I think the red color corresponds to the rewards, and the blue color corresponds to the success rate. However, it seems that there are four environments as shown in the legend, regardless of the color.

[2] Newman, Mark EJ. "Efficient computation of rankings from pairwise comparisons." Journal of Machine Learning Research 24.238 (2023): 1-25.

**Ethical Concerns:**

["NO or VERY MINOR ethics concerns only"]

**Final Justification:**

While the authors address some of my concerns, I still have some reservations regarding the novelty, theoretical analysis, and certain potentially incorrect claims, as partly acknowledged by the authors. After reviewing the other reviewers' comments, I have decided to maintain my current score. However, I am not opposed to acceptance given the merits of the paper.

**Quality:**

3

**Strengths And Weaknesses:**

**Strength**

1. The paper is well-written and easy to follow.
2. The experiments are extensive in both online and offline settings, together with some ablation studies.

**Weakness**
1. The innovation is limited. The contribution lies primarily in applying the established concepts (i.e., margin regularization and conservative Q-value estimation) to the PbRL setting rather than introducing fundamentally new algorithms or theoretical frameworks.

2. Although the paper emphasizes the efficiency of STAR in limited feedback, it lacks thorough ablation studies analyzing how performance changes with varying numbers of preference-labeled trajectories. Ablation studies on the number of trajectories would be beneficial.

3. Although the paper includes some theoretical analysis on conservative Q-value estimation, this aspect has already been extensively studied in prior works like [1], offering not too much novelty. More critically, the paper lacks theoretical guarantees specific to the PbRL setting. In particular, it is not evident whether the objective with the introduced margin can ensure the optimality of the learned reward. And it does not provide formal analysis or generalization bounds that account for the limited preference feedback.

[1] Kumar, Aviral, et al. "Conservative q-learning for offline reinforcement learning." Advances in neural information processing systems 33 (2020): 1179-1191.

---

> ### Author Rebuttal · Authors · 2025-07-31
>
> We sincerely thank Reviewer vhLh for the detailed and critical feedback. We appreciate the acknowledgment of our paper's **clarity and extensive experiments**. The concerns raised about novelty, theoretical guarantees, and the ablation study on feedback are crucial, and they have prompted us to conduct new experiments and significantly clarify the positioning and contributions of our work. We provide detailed point-by-point responses below and believe the resulting discussion strengthens our paper considerably.
>
> **Q1: Reguarding ablation on feedback quantity (Weaknesses #2).**
> > **A1**: We have discussed the effect of feedback quantity on performance of STAR in Fig.6. To further validate STAR's feedback efficiency, we conduct the extensive ablation study on the impact of feedback quantity on task `Window Open`. The results across multiple tasks are presented below, which are averaged over five seeds, and will be added to the paper.
> >
> >Table 1. Walker Walk (measured by episode return, results in Fig.6 of our origial manuscript) .
> >
> >|Method| 100 | 200 | 500 | 1000|
> >| -----| --- | --- | --- | --- |
> >|PEBBLE|612±87|766±78|814±45|954±32|
> >| SURF |497±38|540±34|768±27|949±25|
> >| MRN  |695±33|831±25|884±24|956±25|
> >| STAR |752±31|839±24|936±22|960±21|
> >
> >Table 2. Window Open (measured by success rate).
> >
> >|Method| 100 | 500 | 1000| 2000|
> >| -----| --- | --- | --- | --- |
> >|PEBBLE|10±6 |31±33|55±42|80±24|
> >| SURF |20±20|32±25|42±26|85±13|
> >| MRN  |27±23|47±21|89±9|96±6|
> >| STAR |65±11|76±12|96±6|98±4|
> >
> >These results clearly show that STAR's performance advantage is most pronounced in the critical low-feedback regime. For instance, in Window Open with only 100 preferences, STAR achieves a 65% success rate, more than doubling the performance of the next best method (MRN at 27%). This directly and empirically substantiates our central claim of superior feedback efficiency. As feedback increases, all methods improve, but STAR consistently learns faster and more stably. We will add this crucial study to our appendix.
>
>
> **Q2. Regarding novelty and theory (Weaknesses #1, #3).**
> > **A2**: We thank the reviewer for these critical questions on novelty and theoretical guarantees. We would like to position our contribution more clearly.
> > 1. **On Novelty: A Synergistic Solution to a Dual Problem.** We agree that the core components (margin concepts, conservative Q-learning) are not new in isolation. Our primary contribution is not in inventing these primitives, but in:
> First, identifying and diagnosing the dual failure modes in feedback-limited PbRL: (1) reward model overfitting and (2) policy overestimation exploiting that flawed reward. Prior work has often addressed only one of these.
> Second, proposing a novel and synergistic synthesis (STAR) that holistically addresses both issues. The novelty lies in the insight that these two problems are coupled and require a coupled solution. PMR creates a more robust reward signal, and our policy regularization ensures the agent does not exploit the residual uncertainty in that signal. This specific, motivated combination for PbRL is new and, as our experiments show, highly effective.
> 2. **On Theoretical Guarantees for PbRL.**
> We agree with the reviewer on both points and wish to clarify the scope of our analysis.\
> **(1) Conservative Q-Estimation**: Our Theorem 1 is not intended to be a novel theoretical result on its own. As the reviewer correctly notes, the properties of conservative updates are well-studied [1]. The purpose of Theorem 1 is to provide a formal justification within our paper's context, showing that our specific in-distribution bootstrapping mechanism indeed yields a conservative value function, thereby grounding its use as a regularizer for the policy.\
> **(2) Guarantees for PMR and Generalization**: Providing formal generalization bounds for a complex system coupling preference learning and deep RL is a widely acknowledged open problem, and such an analysis is beyond the scope of this primarily empirical and algorithmic work. However, our motivation for PMR is theoretically principled. Standard likelihood maximization (Eq. 2) presents an ill-posed, unbounded optimization target for a neural network approximator (more on this in Q3). PMR replaces this with a well-posed, bounded target ($log((1-ε)/ε)$). While not a proof of reward optimality, this is a principled way to improve generalization and prevent overfitting, which is a key theoretical motivation for its use.\
> We will revise the manuscript to state this positioning more explicitly, emphasizing that our contribution is a novel algorithmic framework for PbRL, supported by strong empirical results and justified by established theoretical principles.
>
> **Q3. The "Infinite Reward Difference" claim in Eq.(2) (Questions #1).**
> > **A3**: Thank you for this very sharp and important question. In the classical statistical setting described by the Bradley-Terry model (as in Newman [2]), for a given finite dataset with a strongly connected comparison graph, the global MLE indeed corresponds to a set of finite score values.
> >
> > In this deep learning context, the issue arises from the nature of the loss function for each training sample. The standard cross-entropy loss for a single preference pair $(\sigma_0,\sigma_1)$ is equivalent to maximizing $\log P[\sigma_0>\sigma_1]=\log(\text{Sigmoid}(\hat{r}(\sigma_0)−\hat{r}(\sigma_1)))$. This loss is only minimized as the logit—the reward difference $\Delta \hat{r}=\hat{r}(\sigma_0)−\hat{r}(\sigma_1)$—approaches infinity. A powerful neural network has the capacity to continuously push $\Delta \hat{r}$ higher for the pairs in its training batches, leading to an over-confident model that fits the training data perfectly but generalizes poorly. This is the "reward over-optimization" problem we address. Therefore, our claim is not about the existence of a global finite MLE on a static dataset, but about the ill-posed nature of the optimization target for a neural network approximator, which drives it towards unbounded differences. Our PMR directly tackles this by replacing the unbounded target with a principled, finite margin, thus making the learning problem well-posed and preventing overfitting. We sincerely apologize that this critical distinction was not made clear in our manuscript. We will revise Section 3.1 to explicitly state this, clarifying that our analysis pertains to the optimization of neural network-based reward models in deep PbRL, and discuss the connection to the classical view.
>
> **Q4. The "Infinite Reward Difference" claim in Eq.(2) (Questions #1).**
> > **A4**: Thank you for this sharp and important question. Your point is correct and gets to the heart of a subtle but critical distinction we failed to make clear in the paper. In the classical setting of the Bradley-Terry model on a static, connected dataset (as in Newman [2]), the MLE corresponds to finite scores.
> >
> > And our claim refers specifically to the optimization dynamics of a high-capacity neural network used as a reward function approximator. In this setting, the key issue is that the standard cross-entropy loss (Eq.(2)) drives the optimization towards an unbounded reward difference. As you insightfully noted, the objective "wants" the reward difference to be infinite because that is the only way for the loss on a given sample to approach zero.
> > While the reward values do not become literally infinite in practice, this unbounded optimization tendency is precisely what we define as an "ill-posed" objective. It encourages the model to become over-confident in its predictions for training data, leading to poor generalization and the reward over-optimization problem we aim to solve.
> > Our Preference Margin Regularization (PMR) directly addresses this by replacing the unbounded target with a principled, finite margin. This regularizes the model by giving it a well-posed, achievable target, thus preventing it from overfitting to the training preferences.
> >
> >We are grateful for this opportunity to clarify. We will revise Section 3.1 to explicitly draw this distinction between the classical statistical view and the deep learning optimization view, and incorporate this more precise explanation.
>
> **Q5. Clarification on Figure 5(a) (Questions #2)**
> > **A5**: We sincerely apologize for the confusion caused by Figure 5(a). The presentation was indeed unclear. Your initial interpretation was on the right track. The figure plots performance for four distinct tasks, but uses two different metrics mapped to two y-axes:\
> **Red lines (left y-axis)**: Represent tasks measured by episode return (Cheetah Run, Quadruped Walk).\
> **Blue lines (right y-axis)**: Represent tasks measured by success rate (Button Press, Sweep Into).
> >
> >This design was chosen to compactly display results but clearly sacrificed clarity. We will completely redesign this figure in the revised manuscript to ensure there is no ambiguity. Thank you for pointing this out.
>
>
> **References:**\
> [1] Kumar, Aviral, et al. "Conservative q-learning for offline reinforcement learning." Advances in neural information processing systems 33 (2020): 1179-1191.\
> [2] Newman, Mark EJ. "Efficient computation of rankings from pairwise comparisons." Journal of Machine Learning Research 24.238 (2023): 1-25.

---

> ### Comment · Reviewer_vhLh · 2025-08-05
>
> Thanks to the authors for their detailed response and valuable insights. Some of my concerns have been effectively addressed (e.g., A1). However, I still have reservations regarding the novelty, theoretical analysis, and certain potentially incorrect claims, as partly acknowledged by the authors. After reviewing the other reviewers' comments, I have decided to maintain my current score. However, I am not opposed to acceptance given the merits of the paper.

---

> > ### Author Response · Authors · 2025-08-07
> >
> > Dear Reviewer vhLh,
> >
> > Thank you for your follow-up and for acknowledging our efforts to address your concerns. We appreciate the opportunity to clarify our work further and address your reservations regarding our claims, novelty, and theoretical analysis.
> >
> > 1. **Regarding the Claim of "Infinite Reward Difference".**
> > >We agree with the conclusion from [1] that the likelihood objective has a global maximum with a finite reward estimate. Our original phrasing—"the optimization objective is toward assigning infinite reward differences"—was an attempt to describe the incentive provided by the loss function (Eq.(2)), not the final output of the reward model.
> > > \
> > >The standard cross-entropy loss for a preference pair ($\sigma^0$,$\sigma^1$), where $\sigma^0$ is preferred, is $L = − \log( \text{sigmoid}(\Delta \hat{r}))$ , with $\Delta \hat{r}$ being the difference in summed rewards. Mathematically, this loss approaches its theoretical minimum of 0 only as $\Delta \hat{r}\to\infty$ . In practice, $\Delta \hat{r}$ will be finite. However, as long as $\Delta \hat{r}$  is not infinite, the loss is not minimized, and a non-zero gradient will continue to push the reward optimization. This persistent pressure to widen the reward difference for the few available preference labels is what leads to "reward over-optimization"—the model becomes overly confident and specialized to the training data, harming generalization.
> > > \
> > >This is precisely the problem Preference Margin Regularization is designed to solve. PMR introduces a finite, reachable optimization target. By setting the derivative of the PMR loss to zero, we show that the optimal reward difference becomes a finite value: $\log(\frac{1-\epsilon}{\epsilon})$. Once the reward difference reaches this margin, the optimization objective is satisfied. This prevents the model from needlessly exaggerating reward differences and thus mitigates overfitting to the limited feedback.
> > > \
> > > We will revise this section in our paper to state this argument with greater clarity and precision.
> >
> > 2. **On the Novelty and Contribution of STAR.**
> > >We respectfully wish to clarify our view on the novelty of STAR. Our primary contribution is not the invention of a new, standalone conservative Q-learning method. Instead of, our core novelty and contribution lies in the identification of the inefficiency in the PbRL learning loop and providing holistic solution and pratical algorithm.
> >
> > We acknowledge that our paper does not provide a formal proof that the reward function learned via PMR retains optimality, nor does it establish new generalization bounds for PbRL under limited feedback. We believe these are excellent and highly non-trivial open questions that are central to the field's advancement, and we see them as valuable directions for future work.
> >
> > Thank you once again for your rigorous and constructive review. Your feedback has been instrumental in helping us improve the clarity of our paper. We are confident that by incorporating these clarifications, the final manuscript will be substantially stronger and will more accurately represent our contributions.
> >
> >
> > **Reference:**\
> > [1] Newman, Mark EJ. "Efficient computation of rankings from pairwise comparisons." Journal of Machine Learning Research 24.238 (2023): 1-25.

---

> > ### Author Response · Authors · 2025-08-09
> >
> > Dear Reviewer vhLh,
> >
> > We hope this message finds you well.
> >
> > With the discussion period nearing its end, we wanted to briefly and respectfully follow up. We were hoping to hear whether our latest response successfully clarified your remaining reservations.
> >
> > Your expert feedback has been invaluable. Given your comment that you are "not opposed to acceptance given the merits of the paper," your support would be crucial for us.
> >
> > Thank you once again for your time and rigorous review.

---

### Official Review · Reviewer_Q2Zo · 2025-07-03

**Clarity:** 3
**Significance:** 3
**Originality:** 3
**Rating:** 5
**Confidence:** 4

**Summary:**

Preference Based Reinforcement learning allows for learning without explicit reward signal; however gathering these preferences may be expensive. When only a small number of preferences are available, it is very easy for the learned reward model to overfit. Moreover, Q value learning also suffers from over-estimation when being bootstrapped with limited amount of off-policy data. The paper introduces two complementary regularization techniques for these pain-points.

(1) Regularization of the reward model to be smooth adn creates a bounded margin between preferred/rejected samples ; thus preventing the reward model form becoming overconfident.

(2) Regularization of the policy to be close to a reference policy which is obtained by bootstrapping from a "pessimistic" estimate of Q value. This pessimistic estimate is obtained by only consider in-distribution actions from the replay buffer;

The proposed approach consistently outperforms the baselines by a decent margin in both offline and online settings while not adding any significant computation overhead.

**Questions:**

- How was the subset of environments form D4RL chosen for the offline setting experiments.
- Equation 9 uses Q_ξ(s_{t+1}, a_{t+1}). However in first glance it seems like it should be a sample from the empirical behavioral distribution of the dataset. If it is so may be it is worthwhile pointing that it should be sampled from μ_empirical(·|s_{t+1}) and directly taking it from the transition is an approximation.
- Do the authors think that regularization will be more or less helpful when the preferences are noisy? it could be interesting to see how the regularization helps or suffers when the noise is added systematically in the preferences.

**Ethical Concerns:**

["NO or VERY MINOR ethics concerns only"]

**Final Justification:**

I am inclined to accept. The problem of being able to learn from preference directly has extensive applications. The regularization techniques have shown to be complementary to each other and provides meaningful improvement. Moreover. authors have also provided evidence that it may be useful / complementary to other offline RL policy regularization as well.

**Limitations:**

Yes

**Quality:**

3

**Strengths And Weaknesses:**

*Strengths:*
- The proposed regularizations in their approach(STAR) are well motivated for the problem space and has been experimentally shown to be complementary through the ablation studies.
- STAR outperforms the baselines across synthetic offline datasets, online environments as well as human preference settings.
- The authors provide theoretic guarrantees that the conservative Q-function lowerbounds the optimal Q function in tabular settings.
- Experiments on quality of learned reward, value estimation accuracy and effect of feedback quantities were very welcome additions.
- The work overall takes a step towards making PbRL more practical for smaller preference budgets.

*Weakness:*
- The "pessimism" of the Q-function greatly depends on the quality of the offline dataset, as it is with respect to the behavioral policy. The dataset which is anchored to preferences may not always consist of optimal or near optimal rollouts. It could just be Very bad and Very good rollouts. It would have been nice to see experiments with a controlled set of datasets with different levels of diversity of the trajectories (in terms of cumulative rewards) in is used for bootstrapping.
- Since the regularizations are both independently motivated, it could be interesting to see the performance of IQL with learned reward function with and without PMR. Comparing the performance of IQL + PMR with STAR could reveal more about the quality of policy regularization in STAR.

---

> ### Author Rebuttal · Authors · 2025-07-31
>
> We sincerely thank Reviewer Q2Zo for the insightful and constructive feedback. The suggestions to investigate the impact of each component in offline settings and to assess robustness to noisy feedback are particularly valuable and have significantly helped us improve the paper. In response, we have conducted additional experiments and provide detailed, point-by-point replies below. We believe these additions further reinforce the contributions of our work.
>
> **Q1. Regarding the impact of offline dataset quality on STAR’s performance (Weakness #1).**
> > **A1**: Thank you for this excellent point. We agree that this is a critical aspect, especially in the offline setting. We address this concern from two perspectives, aligning with our online and offline experiments:\
> **(1) Online Setting**: The issue of a static, low-quality dataset is naturally mitigated. The replay buffer is continuously populated with new trajectories from the learning agent. As cited in our paper [1, 2], PbRL algorithm typically employ an unsupervised exploration at the very begining to encourage a diverse set of trajectories enters the buffer. Thus, the growing dataset provides a rich empirical distribution for $Q_\xi$ to learn from, preventing it from being anchored to a narrow or polarized subset of behaviors.\
> **(2) Offline Setting**: This is where your concern is most critical. We intentionally evaluated STAR on D4RL datasets with varying quality and diversity to address this (i.e., expert, replay, diverse). Our strong performance across these varied datasets (as shown in our original manuscript) demonstrates that our policy regularization is effective even when the behavior policy is not unimodal or near-optimal. We will clarify this aspect in our experiment section to highlight how our dataset choices directly test this robustness.
>
> **Q2. Regarding the effect of two components of STAR on offline settings (Weakness #2).**
> > **A2**: Thank you for this excellent suggestion to further disentangle the effects of our two components. Following your advice, we conducted a new ablation study in the offline setting, comparing STAR against IQL using a reward model trained with and without our Preference Margin Regularization (PMR).
> > The results are summarized below:
> > |Method|IQL w/o PMR|IQL w/ PMR|**STAR (Ours)**|
> > | -|-|-|-|
> > |antmaze-large-diverse-v2|18.8|31.5|58.7|
> > |hopper-medium-expert-v2 |63.8|78.1|96.8|
> > |lift-ph|89.2|93.7|98.0|
> >
> >These results lead to two clear and important conclusions: \
> >**(1) PMR is a Generally Effective Component**: Adding PMR to the reward learning process significantly boosts the performance of a strong baseline like IQL (e.g., 18.8 -> 31.5 on AntMaze). This confirms that PMR effectively mitigates reward overfitting and is a valuable contribution in its own right. \
> >**(2) Policy Regularization Provides Crucial, Synergistic Gains**: Even when IQL is augmented with the improved PMR-trained reward model, STAR still substantially outperforms it across all tasks. This decisively demonstrates that our policy regularization provides a critical advantage beyond simply using a better reward function. It effectively tackles the value overestimation problem that persists even with a regularized reward, confirming the synergistic and significant contribution of our full STAR algorithm.
> >
> >We will add this valuable ablation study to the appendix of our paper to make these points clear.
>
>
> **Q3. Selection criteria for D4RL benchmark environments (Questions #1).**
> > **A3**: We selected our D4RL environments based on a principled approach aimed at ensuring a rigorous, comprehensive, and fair evaluation. Our criteria were:\
> **(1) Comparability**: We chose environments (e.g., AntMaze, Hopper) that are standard benchmarks in both offline RL and recent offline PbRL works [3, 4], ensuring our results are directly comparable to the state-of-the-art.\
> **(2) Task Diversity**: The selected tasks span a wide range of challenges, including navigation (AntMaze), locomotion (Hopper), and manipulation (Lift), to demonstrate the general applicability of STAR.\
> **(3) Data Quality Diversity**: Crucially, and connecting to your first question (Q1), we deliberately selected datasets with varying quality levels (replay, expert, diverse). This was essential to test STAR's robustness and its core ability to learn effectively from the kind of suboptimal, mixed-quality data that is common in real-world offline scenarios. This principled selection ensures that our claims are supported by evidence from a representative set of challenging and diverse offline settings.
>
>
> **Q4. Clarify using an approximation in Eq.(9) (Questions #2).**
> > **A4**: Thank you for this sharp and accurate observation. You are correct. Using the single next action $a_{t+1}$ from the transition tuple $(s_t, a_t, s_{t+1}, a_{t+1})$ in Eq.(9) is a practical, one-sample Monte Carlo approximation of the expectation over the dataset's empirical policy $μ(·|s_{t+1})$. This is a standard and computationally efficient practice in the field, adopted by prominent offline RL algorithms like CQL [5] and IQL [6], as it avoids the complexity and potential estimation errors of explicitly modeling the behavior policy. To improve the paper's precision, we will add a sentence in Section 3.2 to explicitly clarify this point as you've suggested.
>
> **Q5. Regarding robustness to noisy feedback (Question #3)**
> > **A5**: This is a very important question. While robustness to label noise was not the primary focus of our work, we agree it's a key challenge for real-world PbRL. To explore it, we conduct experiments on `Cheetah Run` and `Button Press` with ratios of noisy data as {0\%, 5\% 10\%, 15\%, 20\%, 25\%}. We systematically injected label noise (flipping preferences) and compared STAR with our backbone (PEBBLE).
> >
> >Table 1. Performance of STAR on Cheetah Run with various level noise.
> >|Method/Noise|0\%|5\%|10\%|15\%|20\%|
> >|-|-|-|-|-|-|
> >|PEBBLE|612|423|/|/|/|
> >| STAR |713|702|629|455|/|
> >
> >Table 2. Performance of STAR on Button Press with various level noise.
> >
> >|Method/Noise|0\%|5\%|10\%|15\%|20\%|25\%|
> >|-|-|-|-|-|-|-|
> >|PEBBLE|23\%|/|/|/|/|/|
> >| STAR |95\%|80\%|67\%|67\%|40\%|/|
> >
> >These results demonstrate that STAR is more robust when the preferences are noisy. While both methods degrade as noise increases, STAR consistently and significantly outperforms PEBBLE. For example, at 15% noise, STAR maintains a strong performance of 67\%, whereas PEBBLE's performance has dropped sharply. This demonstrates that STAR's framework indeed provides substantial robustness against noisy human feedback. We will add this experiment and analysis to the appendix.
>
> **References:**\
> [1] Meta-reward-net: Implicitly differentiable reward learning for preference-based reinforcement learning. NeurIPS, 2022.\
> [2] SURF: Semi-supervised reward learning with data augmentation for feedback-efficient preference-based reinforcement learning. ICLR, 2022. \
> [3] Inverse Preference Learning: Preference-based RL without a Reward Function. NeurIPS, 2023.\
> [4] In-dataset trajectory return regularization for offline preference-based reinforcement learning. AAAI, 2025.\
> [5] Conservative q-learning for offline reinforcement learning. NeurIPS, 2020.\
> [6] Offline Reinforcement Learning with Implicit Q-Learning. ICLR, 2022.

---

> > ### Comment · Reviewer_Q2Zo · 2025-08-04
> >
> > Thank you for the rebuttal. The robustness and added PMR ablation experiments definitely helps to understand benefits the approach more intuitively. I maintain my acceptance score as the responses have satisfactorily resolved the key issues raised.
> >
> > Regarding the offline dataset quality, while D4RL definitely has the diversity, it could be interesting to also have an experiment where dataset are gathered using epsilon greedy policies where the epsilon is gradually increased. I am just curious to see how margin regularization performs when margins in the dataset are varied systematically.

---

> > > ### Author Response · Authors · 2025-08-09
> > >
> > > We sincerely thank Reviewer Q2Zo for positive feedback and for acknowledging that our previous rebuttal and experiments have satisfactorily resolved the key issues. We are happy to hear your support for acceptance.
> > >
> > > We also thank you for your insightful suggestion to investigate our method's performance on datasets with systematically varied quality. We agree that this is an interesting experiment. We have conducted this additional analysis, and the results further highlight the benefits of our proposed margin regularization.
> > >
> > > **1) Experimental Setup**
> > > > To address the reviewer's curiosity, we designed the following experiment:
> > > > \
> > > Base Policy: We first trained a proficient policy using the SAC algorithm in the Hopper, Walker2d and HalfCheetah environments.\
> > > Dataset Generation: We then generated 8 distinct offline datasets for each environment using an $\epsilon$-greedy version of this base policy.
> > > Systematic Variation: The level of exploratory noise (and thus the quality and margin distribution of the dataset) was systematically controlled by varying $\epsilon$ across the set: {0, 0.01, 0.05, 0.1, 0.2, 0.3, 0.4, 0.5}. A higher $\epsilon$ corresponds to a dataset with more random actions.
> > > Data Size: Each of the 8 datasets contains 1000 complete episodes.
> > >
> > > **2) Results and Analysis**
> > > >We have conducted the suggested experiment, and the results are presented in the table below. The scores are normalized based on D4RL standards. Our method (STAR) is compared against the Preference Transformer (PT)[1] (offline PbRL baseline).
> > > >| Task/$\epsilon$ |Method|0.0|0.01|0.05|0.1|0.2|0.3|0.4|0.5|
> > > >|-|-|-|-|-|-|-|-|-|-|
> > > >|Hopper|STAR|111.7|111.7|111.4|111.3|103.6|62.8|41.8|20.8|
> > > >||PT|103.2|100.1|93.6|81.2|67.4|37.3|10.5|0.0|
> > > >|Walker2d|STAR|107.1|106.2|102.7|74.3|59.7|37.9|23.8|8.3|
> > > >||PT|92.9|88.1|72.3|20.7|0.0|/|/|/|
> > > >|HalfCheetah|STAR|97.1|96.8|94.2|86.8|79.8|57.8|28.3|13.9|
> > > >||PT|68.9|60.0|51.2|49.7|26.5|0.0|/|/|
> > > >
> > > >\
> > > >The results offer a clear insight: our method's superiority grows as the data quality systematically decreases, directly validating its design for robustness.
> > > >(1) Robustness: While both methods perform well on near-expert data ($\epsilon=0$), the baseline's performance rapidly collapses as stochasticity increases. In stark contrast, our method (STAR) maintains significant performance.
> > > >(2) Core Mechanism Validation: Increasing $\epsilon$ systematically injects more challenging, low-margin, and conflicting preference pairs into the dataset. The baseline's failure suggests it is brittle to this kind of noise. Our method's strong performance is direct evidence that its core mechanism—explicit margin regularization—is precisely what is needed to create a clear and robust decision boundary.
> > >
> > > We hope these additional results provide a more comprehensive understanding of our method's robustness. We thank the reviewer once again for their valuable and constructive engagement with our work.
> > >
> > > **Reference:**\
> > > [1] Preference Transformer: Modeling Human Preferences using Transformers for RL. ICLR, 2023.

---

### Note · Authors · 2025-08-13

We are grateful to the reviewers for their rigorous engagement, which has been instrumental in strengthening our manuscript.

We are pleased the discussion led to a clear consensus on the paper's merits and summarize the key strengths below:
- **Significant & Well-Defined Problem:** Reviewers agreed the problem we address is "interesting and significant" (LJRt), praising our "clear problem formulation" (qN2G) and "well-motivated" approach (Q2Zo).
- **Novel  Solution:** Our algorithm was recognized for its holistic design. Reviewers noted its core techniques are "complementary" (Q2Zo), and new experiments confirmed their "synergy... is critical for... success" (LJRt), forming a novel and effective combination for PbRL.
- **Comprehensive Experiments:** The paper's "extensive experiments" (vhLh) and "strong performance" (qN2G) were highlighted. The work is now validated against recent SOTA methods (QPA, DTR), which "satisfactorily resolved" key concerns about experimental rigor (LJRt).
- **Quality and Responsiveness:** The paper was consistently praised as "well-written" and "easy to follow" (vhLh, qN2G). Crucially, our active engagement in the discussion, leading to a significant score increase from Reviewer LJRt, affirmed both the paper's quality and our responsiveness.

We also want to acknowledge the initial concerns, as addressing them directly has improved our paper:
- Novelty: We addressed this by clarifying that **our contribution lies in the diagnosis of a coupled problem in PbRL and the novel, synergistic synthesis of a holistic solution.** New offline ablation studies provided direct empirical evidence for this crucial synergy.
- Experimental Scope: This was resolved by adding new comparisons and a suite of ablation and sensitivity studies, which thoroughly addressed questions about robustness, hyperparameter complexity.

To enhance quality, the final manuscript will incorporate all improvements. We will integrate all new results, provide a more precise explanation of the "Infinite Reward Difference" claim (Sec 3.1) and redesign Fig 5a as discussed.

Finally, we are excited by the reviewers' recognition of our work's potential. As Reviewer Q2Zo noted, it "takes a step towards making PbRL more practical." We believe our work contributes not only an effective algorithm but also a clear analysis of core challenges in the field. The positive outcome of the review process affirms this work is a valuable contribution ready for publication.

---

### Decision · Program_Chairs · 2025-09-17

**Decision:**

Accept (poster)

**Comment:**

The paper looks at the issue of overfitting in the case of preference based learning. They break down the problem into two aspect: (a) overfitting in the reward-modeling phase, and (b) exaggeration of the q-value over-estimation issues through this during the RL phase. To remedy this, inspired by existing works on margin regularization, authors propose leveraging margin based BCE loss for reward modeling. Further, to remedy the over-estimation bias, authors propose using CQL (Kumar et al. 2020) with the reward model learned with margin regularization.

Reviewers mostly praised the work. Some questions were raised regarding, offline data selection, environment selection, feedback noise, hyper-paramater sensitivity, etc. Authors engaged in a constructive discussion and addressed most of the issues.

In terms of technical novelty, as some reviewers noted, the contributions are modest. However, I commend the authors for systematically studying the problem:

- exposition of the issue and presentation is well done.
- flow of arguments are logical and clear.
- Both aspects of the contribution are evaluated individually + together, and additional hyper-param ablations are presented.
- Several benchmarks and multiple baselines are used.
- I would have appreciated more than 5 trials for at least some of the smaller benchmarks. However, given that over a dozen domains have been used and the trend of the results are in the right direction, I think these results are still useful and will be of interest to the audience.

Minor: I would request the authors to replace the word “dual” with “double” in the title, as “dual regularization” sounds like using dual variables to do some regularization, which is not the case.